# CyberHost: A One-stage Diffusion Framework for Audio-driven Talking Body Generation

**Gaojie Lin**[1*], **Jianwen Jiang**[1*†], **Chao Liang**[1] , **Tianyun Zhong**[2‡], **Jiaqi Yang**[1] , **Yanbo Zheng**[1]
[1]ByteDance, [2]Zhejiang University
{lingaojiecv,jianwen.alan,liangchao.0412}@gmail.com
zhongtianyun@zju.edu.cn

## Abstract

Diffusion-based video generation technology has advanced significantly, catalyzing a proliferation of research in human animation. While breakthroughs have been made in driving human animation through various modalities for portraits, most of current solutions for human body animation still focus on video-driven methods, leaving audio-driven taking body generation relatively underexplored. In this paper, we introduce CyberHost, a one-stage audio-driven talking body generation framework that addresses common synthesis degradations in half-body animation, including hand integrity, identity consistency, and natural motion. CyberHost's key designs are twofold. Firstly, the Region Attention Module (RAM) maintains a set of learnable, implicit, identity-agnostic latent features and combines them with identity-specific local visual features to enhance the synthesis of critical local regions. Secondly, the Human-Prior-Guided Conditions introduce more human structural priors into the model, reducing uncertainty in generated motion patterns and thereby improving the stability of the generated videos. To our knowledge, CyberHost is the first one-stage audio-driven human diffusion model capable of zero-shot video generation for the human body. Extensive experiments demonstrate that CyberHost surpasses previous works in both quantitative and qualitative aspects. CyberHost can also be extended to video-driven and audio-video hybrid-driven scenarios, achieving similarly satisfactory results. Video samples are available at https://cyberhost.github.io/.

## 1 Introduction

Human animation aims to generate realistic and natural human videos from a single image and control signals such as audio, text, and pose sequences. Previous works (Prajwal et al., 2020; Yin et al., 2022; Wang et al., 2021; Ma et al., 2023; Zhang et al., 2023; Chen et al., 2024b; Xu et al., 2024b;a; Tian et al., 2024; Jiang et al., 2024a; Wang et al., 2024a; Jiang et al., 2024b) have primarily focused on generating talking head videos based on varied input modalities. Among these, audio-driven methods have recently attracted significant interest, particularly those employing diffusion models (Tian et al., 2024; Xu et al., 2024a;b). While these methods can yield impressive results, they are specifically tailored for portrait scenarios, making it challenging to extend them to half-body scenarios to achieve audio-driven talking body generation. This is because generating talking body videos involves more intricate human appearance details and complex motion patterns.

On the other end of the spectrum, some recent studies (Karras et al., 2023; Wang et al., 2024b; Hu et al., 2024; Zhang et al., 2024; Xu et al., 2024c; Huang et al., 2024; Corona et al., 2024) focus on tackling video-driven body animation. Unlike audio-driven settings, these methods rely on pose conditions to provide precise, pixel-aligned body structure prior, enabling the modeling and generation of large-scale human movements and fine-grained local details. Even so, accurately generating detailed body parts remains challenging, as shown in Figure 1. Video-driven body animation methods often require additional motion generation modules and retargeting techniques, which limit

---

[*]Equal contribution.

[†]Project lead

[‡]Done during an internship at ByteDance.

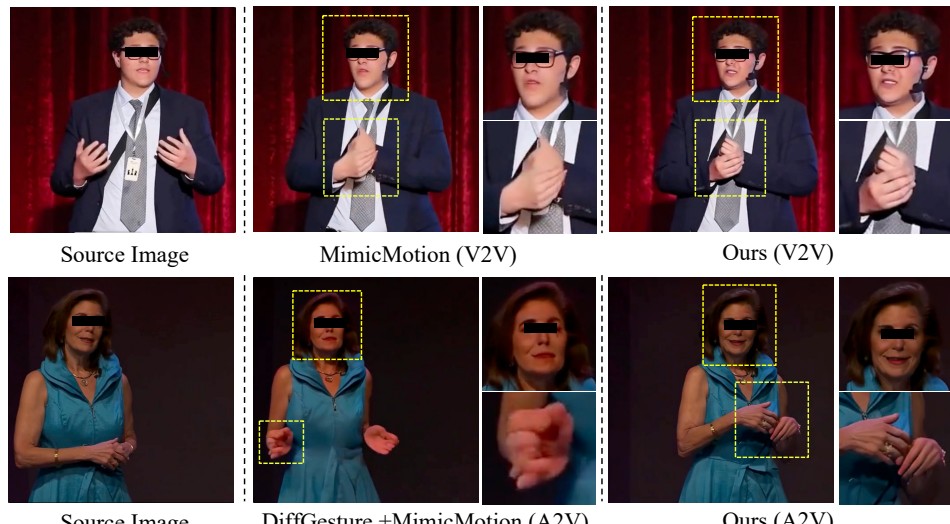

Figure 1: Existing body animation methods struggle to generate detailed hand and facial results in both video-driven (V2V) and audio-driven (A2V) settings. In contrast, our approach ensures hand integrity and facial identity consistency. These differences are also illustrated with videos in the supplementary materials.

their practical applications. Recent works (Liao et al., 2020; Wang et al., 2023c; Hogue et al., 2024; Corona et al., 2024) have explored the implementation of two-stage systems to achieve audio-driven talking body generation. This generally consists of an audio-to-pose module and a pose-to-video module, using poses or meshes as intermediate representations. Nevertheless, this approach faces several critical limitations: (1) The two-stage framework design increases system complexity and reduces the model's learning efficiency. (2) The poses or meshes carry limited information related to expressiveness, constraining the model's ability to capture subtle human nuances. (3) Potential inaccuracies in pose or mesh annotations can diminish the model's performance. Therefore, there is an urgent need to explore how to optimize the generation quality of talking body video within a one-stage audio-driven framework.

In this paper, we aim to address one-stage talking body generation, a topic that remains unexplored in current literature. The challenge lies in two aspects: 1) *Details Underfitting.* Unlike video-driven methods, capturing local structural details from audio signals is difficult, making it harder to ensure the integrity of body parts, such as the face and hands. Moreover, critical human body parts occupy only a small portion of the frame but carry the majority of the identity information and semantic expression. Unfortunately, neural networks often fail to spontaneously prioritize learning in these key regions, intensifying the issue of underfitting in the generation of local details. 2) *Motion Uncertainty.* Unlike portrait animation, body animation encompasses a higher degree of motion freedom and exhibits a weaker correlation between audio cues and limb movement patterns. Consequently, predicting body movements from audio signals introduces a more substantial one-to-many problem, leading to significant uncertainty in the motion generation process. This uncertainty exacerbates instability in the generated talking body videos, thereby complicating the direct adaptation of audio-driven portrait animation techniques to half-body scenarios.

To address these two challenges, we propose CyberHost, a one-stage audio-driven talking body framework capable of zero-shot human videos generation. On one hand, CyberHost introduces a region attention module (RAM) to address the issue of underfitting local details. Specifically, RAM utilizes a learnable spatio-temporal latents bank to capture common local human details from the data, such as topological structures and motion patterns, thereby ensuring the maintenance of structural details. Additionally, it integrates appearance features from local cropped images, serving as identity descriptors, to supplement the identity-specific texture details. On the other hand, to address the motion uncertainty problem, we designed human-prior-guided conditions to incorporate motion pattern constraints and human structural priors into the human video generation process. Specifically, for global motion, we propose a body movement map to constrain the motion space of the human root node, and for local motion, we introduce a hand clarity score to mitigate hand degradation caused by motion blur. Additionally, for human structure, we utilize the skeleton map

of the reference image to extract pose-aligned reference features, thereby providing the model with initial pose information from the reference image.

In our experiments, we validated the effectiveness of the region attention modules and human-prior-guided conditions, Both qualitative and quantitative experiments demonstrate that CyberHost achieves superior results compared to existing methods. Moreover, we validated the exceptional performance of CyberHost in various settings, including audio-driven, video-driven, and multimodal-driven scenarios, as well as its zero-shot video generation capability for open-set images.

We summarize our technical contributions as follows: 1) We propose the first **one-stage audio-driven talking body framework** enabling zero-shot human body animation without relying any intermediate representations, and validate its effectiveness across multiple scenarios. 2) We crafted region attention module (RAM) to enhance the generation quality of key local regions such as hands and faces, by including a spatio-temporal latents bank to learn shared local structural details and an identity descriptor to supplement ID-specific texture details. 3) We designed a suite of human-prior-guided conditions to mitigate the instability caused by motion uncertainty in audio-driven settings.

## 2 RELATED WORK

**Video Generation.** Benefiting from the advancements in diffusion models, video generation has made significant progress in recent years. Some early works (Singer et al., 2022; Blattmann et al., 2023a; Zhou et al., 2022; He et al., 2022; Wang et al., 2023a) have attempted to directly extend the 2D U-Net pretrained on text-to-image tasks into 3D to generate continuous video segments. AnimateDiff (Guo et al., 2024) trained a pluggable temporal module on large-scale video data, allowing easy application to other text-to-image backbones and enabling text-to-video generation with minimal fine-tuning. For controllability, (Wang et al., 2023b) trained a Composer Fusion Encoder to integrate multiple modalities of input as control conditions, thereby making video generation for complex scenes such as human bodies more controllable.

**Body Animation.** Existing body animation approaches mainly (Hu et al., 2024; Wang et al., 2024b; Xu et al., 2024c; Karras et al., 2023; Zhou et al., 2022) focus on video-driven settings, where control signals are pose sequence extract from the driving video. DreamPose (Karras et al., 2023) uses DensePose (Güler et al., 2018) to train a diffusion model for sequential pose transfer. MagicAnimate (Xu et al., 2024c) extends the 2D U-Net to 3D to enhance the temporal smoothness. AnimateAnyone (Hu et al., 2024) employs a dual U-Net architecture to maintain consistency between the generated video and the reference images. Some speech-driven body animation works (Liao et al., 2020; Ginosar et al., 2019; Wang et al., 2023c; Corona et al., 2024) do exist, but they typically employ a two-stage framework. Speech2Gesture (Ginosar et al., 2019) first predicts gesture sequence and then utilizes a pre-trained GAN to render it to final video. Similarly, Vlogger (Corona et al., 2024) employs two diffusion models to separately perform audio-to-mesh and mesh-to-video mapping. Two-stage methods rely on explicit intermediate representations to mitigate the training difficulty in audio-driven settings. However, the limited expressive capabilities of these intermediate representations can also constrain the overall performance. Unfortunately, the end-to-end training of a one-stage diffusion model for audio-driven talking body generation remains unexplored.

## 3 METHOD

### 3.1 OVERVIEW

We develop our algorithm based on the Latent Diffusion Model (LDM) (Blattmann et al., 2023b), which utilizes a Variational Autoencoder (VAE) Encoder (Kingma & Welling, 2014) $\mathcal{E}$ to transform the image $I$ from pixel space into a more compact latent space, represented as $z_0 = \mathcal{E}(I)$, to reduce the computational load. During training, random noise is iteratively added to $z_0$ at various timesteps $t \in [1, ..., T]$, ensuring that $z_T \sim \mathcal{N}(0, 1)$. The training objective of LDM is to predict the added noise at every timestep $t$:

$$L = \mathbb{E}_{z_t, t, c, \epsilon \sim \mathcal{N}(0,1)} \left[ \| \epsilon - \epsilon_\theta(z_t, t, c) \|_2^2 \right], \tag{1}$$

where $\epsilon_\theta$ denotes the trainable components such as the Denoising U-Net, and $c$ represents the conditional inputs like audio or text. During inference, the trained model is used to iteratively remove

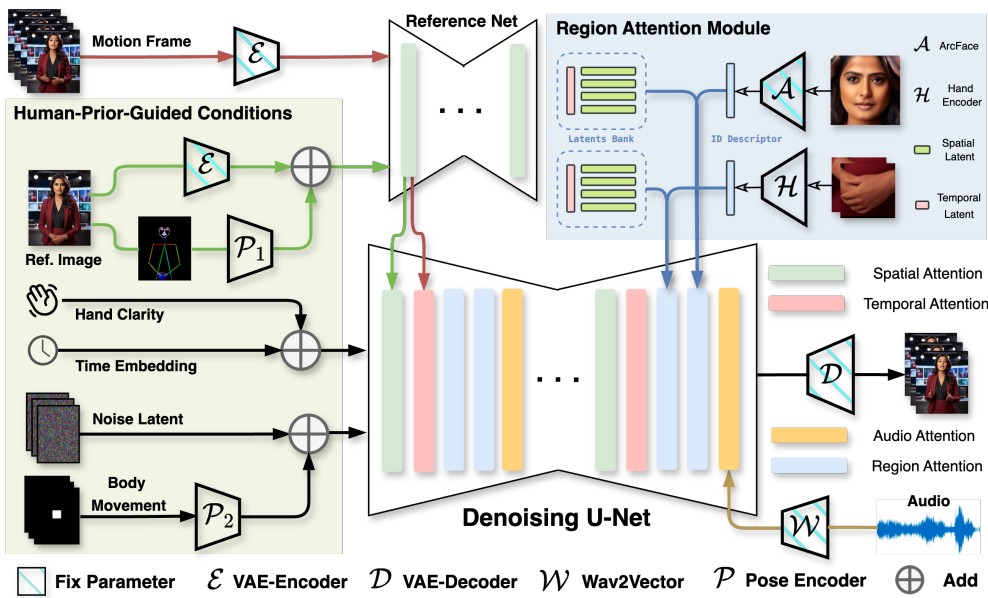

Figure 2: **The overall structure of CyberHost.** We aim to generate a video clip by driving a reference image based on an audio signal. Region attention modules (RAMs) are inserted at multiple stages of the denoising U-Net for fine-grained modeling of local regions. Additionally, Human-Prior-Guided Conditions, including the body movement map, hand clarity score and pose-aligned reference features are also introduced to reduce motion uncertainty. The reference network extracts motion cues from motion frames for temporal continuation.

noise from a noised latent sampled from a Gaussian distribution. Subsequently, the denoised latent is decoded into an image using the VAE Decoder $\mathcal{D}$.

Our proposed CyperHost takes a human reference image and a speech audio clip as inputs, ultimately generating a synchronized human video. The overall architecture is illustrated in Figure 2. We referenced the design of the reference net from AnimateAnyone(Hu et al., 2024) and TryOnDiffusion(Zhu et al., 2023b), as well as the motion frames from Diffused-Heads(Stypulkowski et al., 2024) and EMO (Tian et al., 2024), to construct a baseline framework. Specifically, a copy of the 2D U-Net is utilized as a reference net to extract reference features from the reference image and motion features from the motion frames. For audio, we use Wav2vec (Schneider et al., 2019) to extract multi-scale features. For the denoising U-Net, we extend the 2D version to 3D by integrating the pretrained temporal module from AnimateDiff (Guo et al., 2024), enabling it to predict human body video clips. The reference, motion frames, and audio features are fed into the denoising U-Net in each resblock. They are respectively combined with latent features in the spatial dimension to share the self-attention layer, in the temporal dimension to share the temporal module, and through an additional cross-attention layer to achieve this.

As shown in Figure 2, based on the baseline, we proposed two key designs to address the inherent challenges of the audio-driven talking body generation task. First, to enhance the model's ability to capture details in critical human region, *i.e.*, hands and faces, we adapt the proposed region attention module (RAM), detailed in section 3.2) to both the facial and hand regions and insert them into multiple stages of the Denoising U-Net. RAM consists of two parts: the spatio-temporal region latents bank learned from the data and the identity descriptor extracted from cropped local images. Second, to reduce the motion uncertainty in half-body animation driven solely by audio, several conditions (detailed in section 3.3) have been designed to integrate global-local motion constraints and human structural priors: (1) The body movement map is employed to stabilize the root movements of the body. It is encoded and merged with the noised latent, serving as the input for the denoising U-Net. (2) The hand clarity score is designed to prevent hand prediction degradation caused by motion blur in the training data. It is incorporated as a residual into the time embedding. (3) The pose encoder encodes the reference skeleton map, which is then integrated into the reference latent, yielding a pose-aligned reference feature. Note that the pose encoder for the body movement map and the reference skeleton map share the same model architecture, except for the first convolution layer, but do not share any model parameter.

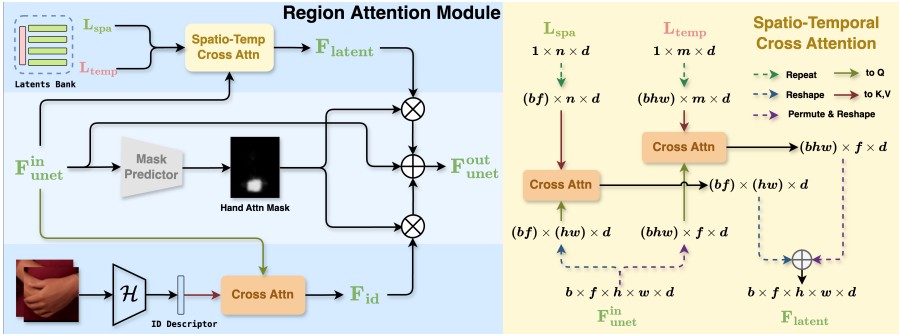

Figure 3: An illustration of region attention module (RAM), using the hand region as an example.

## 3.2 REGION SYNTHESIS WITH REGION ATTENTION MODULES

While the popular dual U-Net architecture effectively maintains overall visual consistency between the generated video and reference image, it struggles with generating fine-grained texture details and complex motion patterns in local areas like the face and hands. This challenge is further exacerbated in the task of audio-driven human body animation due to the absence of explicit control signals. To address this issue, we meticulously designed the structure of the RAM to enhance its ability to learn local details. As shown in Figure 3, our proposed RAM comprises two key parts: spatio-temporal region latents bank and identity descriptor. The former aims to learn identity-agnostic general features, while the latter focuses on extracting identity-specific unique features. Together, they enhance the synthesis of local human regions. In subsequent sections, we apply it to the face and hands, two areas that typically present significant challenges, and confirm its effectiveness.

**Region Latents Bank.** RAM enhances the model by adding a spatio-temporal latents bank as additional training parameters, prompting it to learn shared local structural priors, including common topological structures and motion patterns. The region latents are composed of two sets of learnable basis vectors: $\mathbf{L}_{\text{spa}} \in \mathbb{R}^{1 \times n \times d}$ for spatial features and $\mathbf{L}_{\text{temp}} \in \mathbb{R}^{1 \times m \times d}$ for temporal features, where $n$ and $m$ denote the number of basis vectors and $d$ denotes the channel dimension. We consider the combination of $\mathbf{L}_{\text{spa}}$ and $\mathbf{L}_{\text{temp}}$ as a pseudo 3D latents bank, endowing it with the capability to learn spatio-temporal features jointly. This capability facilitates the modeling of 3D characteristics such as hand motion. Furthermore, we constrain the basis vectors of the latents bank to be mutually orthogonal to maximize its learning capacity.

The regional latent features are integrated into the U-Net through a spatio-temporal cross-attention as shown in Figure 3. Given the backbone feature $\mathbf{F}_{\text{unet}}^{\text{in}}$ from U-Net, we apply cross attention with $\mathbf{L}_{\text{spa}}$ in the spatial dimension and with $\mathbf{L}_{\text{temp}}$ in the temporal dimension. The final output $\mathbf{F}_{\text{latent}}$ is formulated as the sum of two attentions' result,

$$\mathbf{F}_{\text{latent}} = \text{Attn}(\mathbf{F}_{\text{unet}}^{\text{in}}, \mathbf{L}_{\text{spa}}, \mathbf{L}_{\text{spa}}) + \text{Attn}(\mathbf{F}_{\text{unet}}^{\text{in}}, \mathbf{L}_{\text{temp}}, \mathbf{L}_{\text{temp}}) \tag{2}$$

$$= \text{softmax}\left(\frac{\mathbf{Q}\mathbf{K}_{\text{spa}}^T}{\sqrt{d}}\right) \cdot \mathbf{V}_{\text{spa}} + \text{softmax}\left(\frac{\mathbf{Q}\mathbf{K}_{\text{temp}}^T}{\sqrt{d}}\right) \cdot \mathbf{V}_{\text{temp}} \tag{3}$$

where $\text{Attn}(*, *, *)$ denotes the cross attention, $\mathbf{Q}$, $\mathbf{K}$ and $\mathbf{V}$ are the query, key, and value, respectively. We aim for $\mathbf{F}_{\text{latent}}$ to fully utilize the spatio-temporal motion priors of the local region learned within the 3D latents bank, refining and guiding the U-Net features through residual addition. Notably, to effectively focus the latents bank on feature learning for the target local region while filtering out gradient information from unrelated areas, we require a regional mask to weight the residual addition process. Due to the absence of prior information on body part positions in the audio-driven scenario, we employ auxiliary convolutional layers as a regional mask predictor. This predictor directly estimates a regional attention mask $\mathbf{M}_{\text{pred}}$ using the U-Net feature $\mathbf{F}_{\text{unet}}^{\text{in}}$. During training, we use region detection boxes to generate supervision signals $\mathbf{M}_{\text{gt}}$ for mask predictor.

**Identity Descriptor.** The process of learning the latents bank is identity-agnostic. It leverages data to learn the shared local structural and motion patterns priors of the human body. However, identity-specific features such as hand size, skin color, and textures are also important and cannot be overlooked. To complement this, we employ a regional image encoder $\mathcal{R}$ to extract identity-aware

regional features from the cropped region image $\mathbf{I}_r$. The extracted feature is referred to as the identity descriptor. We illustrate this process in the left bottom of Figure 3 using the hand as an example.

Combining the identity-independent latents bank and the identity descriptor, the final output $\mathbf{F}_{unet}^{out}$ of the RAM module with locally enhancement features is obtained as follows:

$$\mathbf{F}_{id} = \text{Attn}(\mathbf{F}_{unet}^{in}, \mathcal{R}(\mathbf{I}_r), \mathcal{R}(\mathbf{I}_r)) \tag{4}$$

$$\mathbf{F}_{unet}^{out} = (\mathbf{F}_{latent} + \mathbf{F}_{id}) * \mathbf{M}_{pred} + \mathbf{F}_{unet}^{in} \tag{5}$$

**Enhance Hand and Facial Regions Synthesis with RAM.** Note that both hand and facial features can be divided into identity-independent common structural features and identity-dependent appearance features. Therefore, the design principle of region attention module ensures its applicability to feature modeling for both hand and facial regions. In terms of implementation details, we use a structure similar to the Pose Encoder but with deeper blocks for the Hand Encoder to enhance its feature extraction capability. The images of both hands are individually cropped and resized to a resolution of 128 for feature extraction, and then concatenated to form the hand identity descriptor. For the facial region, we utilize a pre-trained ArcFace (Deng et al., 2019) network for identity feature extraction. Considering the rich details of the face, facial images are resized to a resolution of 256 for feature extraction. For $\mathbf{M}_{gt}$, both hand and facial detect boxes are obtained by the minimal enclosing area defined by their respective key points.

**Local Enhancement Supervision.** The design of the latents bank and the identity descriptor aims to enhance the model's learning capacity. Additionally, we strive to provide stronger supervisory signals for modeling critical local regions based on the local enhancement features from RAM. Firstly, we employ a local reweight strategy to optimize the original training objective. Specifically, we construct a mask $\mathbf{M}_{gt}$ for critical regions such as the face and hands, and use it to reweight the original training loss $L$ by a factor $\alpha$. Moreover, synthesizing hands is more challenging than synthesizing faces due to the high flexibility of fingers. To address this, we add auxiliary losses to further enhance the synthesis of hand regions. Specifically, after each hand RAM, we pass the locally refined features $\mathbf{F}_{unet}^{out}$ through several convolutional layers to predict the hand keypoints' heatmap $\hat{\mathbf{H}}$, aiding the model in better understanding the hand structure. Finally, our local enhancement loss function is formulated as follows:

$$L_{les} = (1 + \alpha * \mathbf{M}_{gt}^{hand} + \alpha * \mathbf{M}_{gt}^{face}) * L + \frac{1}{N}\sum_{i=1}^{N}\|\mathbf{H}_i, \hat{\mathbf{H}}_i\|_2^2 \tag{6}$$

where $\mathbf{H}$ denotes the ground truth keypoints heatmap, and $N$ denotes the number of region attention modules in U-Net. We found that setting $\alpha = 1$ yielded the most stable results. Considering that the signal-to-noise ratio varies at different time steps $t$, we apply the local enhancement loss with a $50\%$ probability only when $t < 500$.

### 3.3 Human-Prior-Guided Conditions

While RAM enhances the synthesis of critical human regions, generating half-body human motion videos with audio as the sole condition can still introduce several artifacts: 1) random global body movements; 2) probabilistic blurring of hand gestures; 3) ambiguous limb structures. The first two artifacts are caused by motion uncertainty in the audio-driven generation of the talking body, while the last artifact is due to the model's insufficient prior knowledge of human structure topology. Recent video generation studies (Podell et al., 2023; Blattmann et al., 2023a) demonstrate that designing condition inputs, such as resolution and cropping parameters, can enhance the model's robustness to varied data and improve output controllability. Inspired by these studies, we introduce human prior information to the model and design the body movement map and hand clarity score conditions to decouple absolute body motion and hand motion blur during training, reducing the uncertainty caused by the weak correlation between audio and body motion. Additionally, we designed the pose-aligned reference feature to help the model better perceive human structure.

**Body Movement Map.** Frequent body movements, including translations, rotations, or those caused by camera movements, are present in the talking body video data, increasing the training difficulty. To address this issue, we introduce a body movement map to serve as a control signal for the movement amplitude of the body root in generated videos. Specifically, we define a rectangular box

representing the motion range of the thorax point over a video segment. To avoid generating body movements that are overly dependent on the input movement map (which can lead to rigid motion during inference), we augment the size of the rectangular box by 100%-150%. The body movement map is down-sampled and encoded through a learnable Pose Encoder and added as a residual to the noised latent. During inference, we input a body movement map of fixed size to ensure the stability of the overall generated results. Due to the augmentation during training, the audio-conditioned model can still generate natural motion, avoiding rigidity.

**Hand Clarity Score.** Due to exposure duration and rapid hand movements, video data often contains blurry hand images, which lose human region structural details, weakening the model's ability to learn hand structures and causing it to generate indistinct hand appearances. Therefore, we introduce a hand clarity score to indicate the clarity of hand regions in the training video frames. This score is used as a conditional input to the denoising U-Net, enhancing the model's robustness to blurry hand data during training and enabling control over the clarity of the hand images during inference. Specifically, for each frame in the training data, we crop the pixel areas of the left and right hands based on key points and resize them to a resolution of $128 \times 128$. We then use the Laplacian operator to calculate the Laplacian standard deviation of the hand image frames. A higher standard deviation typically indicates clearer hand images, and we use this value as the hand clarity score. The hand clarity score is provided to the U-Net model by residually adding it to the time embedding. During inference, a higher clarity score is applied to enhance the generation results for the hands.

**Pose-aligned Reference Feature.** Recent works (Zhu et al., 2023b; Hu et al., 2024; Tian et al., 2024) have utilized reference net to extract and inject appearance features from the reference image, thereby maintaining overall visual consistency between the generated video and the reference image. However, when dealing with challenging pose reference images, the reference net struggles to accurately perceive the initial pose, especially hand gestures, which affects the quality of hand generation in the whole generated video. To address this, we leverage a pretrained model to perform pose recognition on the reference image. The results are encoded as a skeleton map, which is then processed by a feature extractor and added to the original reference features. This ensures that the extracted reference features not only include the appearance information of the human body but also incorporate its topological structure information provided by the expert pose model, improving the model's robustness to the initial pose.

# 4 EXPERIMENTS

## 4.1 IMPLEMENT DETAILS

The training process is divided into two stages. In the first stage, two arbitrary frames from the training videos are sampled as the reference and target frames to learn the human image generation process. Training parameters include those of the reference net, pose encoder, and basic modules within the denoising U-Net. In the second stage, we begin end-to-end training for generating videos from reference image and audio. All the parameters including those of the temporal layers, audio attention layers, and region attention modules are optimized. Each video clip has a length of 12 frames, with the motion frames' length set to 4. In stage one, we trained at a fixed resolution of $640 \times 384$, and in stage two, we trained at a dynamic resolution with an area approximately equal to $640 \times 384$. Each stage is trained with the learning rate set to $1e^{-5}$. We trained our model on 200 hours of video data featuring half-body speech scenarios, sourced from the internet and comprising over 10,000 unique identities. Further details are provided in the supplementary material. For quantitative evaluation, we designated 269 video segments from 119 identities as the test set.

## 4.2 COMPARISONS WITH STATE-OF-THE-ARTS

Although initially designed for the audio-driven talking body generation task, our method can be easily generalized to other applications, such as video-driven body reenactment and audio-driven talking head. This allows us to validate the advanced nature and generalizability of the CyberHost framework against some of the current state-of-the-art methods in related fields. For evaluation metrics, we use Fréchet Inception Distance (FID) to assess the quality of the generated video frames and Fréchet Video Distance (FVD) (Unterthiner et al., 2019) to evaluate the overall coherence of the generated videos. To assess the preservation of facial appearance, we calculate the cosine similarity

Table 1: Quantitative comparison of audio-driven talking body. ∗ denotes evaluate on vlogger test set.

| Methods | SSIM↑ | PSNR↑ | FID↓ | FVD↓ | CSIM↑ | SyncC↑ | SyncD↓ | HKC↑ | HKV↑ |
|---|---|---|---|---|---|---|---|---|---|
| DiffTED | 0.667 | 15.48 | 95.45 | 1185.8 | 0.185 | 0.9259 | 12.543 | 0.769 | - |
| DiffGest.+MimicMo. | 0.656 | 14.97 | 58.95 | 1515.9 | 0.377 | 0.496 | 13.427 | 0.833 | 23.40 |
| CyberHost (A2V-B) | **0.691** | **16.96** | **32.97** | **555.8** | 0.514 | **6.627** | **7.506** | **0.884** | 24.73 |
| Vlogger ∗ | - | - | - | - | 0.470 | 0.601 | 11.132 | 0.923 | 9.84 |
| CyberHost (A2V-B) ∗ | - | - | - | - | 0.522 | 7.897 | 7.532 | 0.907 | 18.75 |
| w/o Latents Bank | 0.687 | 16.53 | 37.80 | 643.9 | 0.523 | 6.384 | 7.719 | 0.859 | 21.35 |
| w/o Temp. Latents | 0.681 | 16.62 | 36.75 | 624.1 | 0.489 | 6.468 | 7.573 | 0.870 | 19.44 |
| w/o ID Descriptor | 0.690 | 16.95 | 35.83 | 582.9 | 0.422 | 6.418 | 7.669 | 0.881 | 22.64 |
| w/o Hand RAM | 0.686 | 16.80 | 37.71 | 625.9 | 0.498 | 6.510 | 7.574 | 0.869 | 22.98 |
| w/o Face RAM | 0.685 | 16.86 | 35.14 | 612.8 | 0.425 | 6.299 | 7.796 | 0.880 | 24.11 |
| w/o Local Enhancement | 0.687 | 16.92 | 35.25 | 581.5 | 0.461 | 6.127 | 7.930 | 0.866 | 21.35 |
| w/o Body Movement | 0.680 | 16.76 | 39.83 | 668.6 | 0.458 | 6.372 | 7.769 | 0.867 | 27.54 |
| w/o Hand Clarity | 0.686 | 16.73 | 37.81 | 643.8 | 0.503 | 6.556 | 7.556 | 0.849 | 33.00 |
| w/o Pose-aligned Ref. | 0.683 | 16.66 | 38.32 | 660.0 | 0.487 | 6.498 | 7.684 | 0.870 | 23.18 |

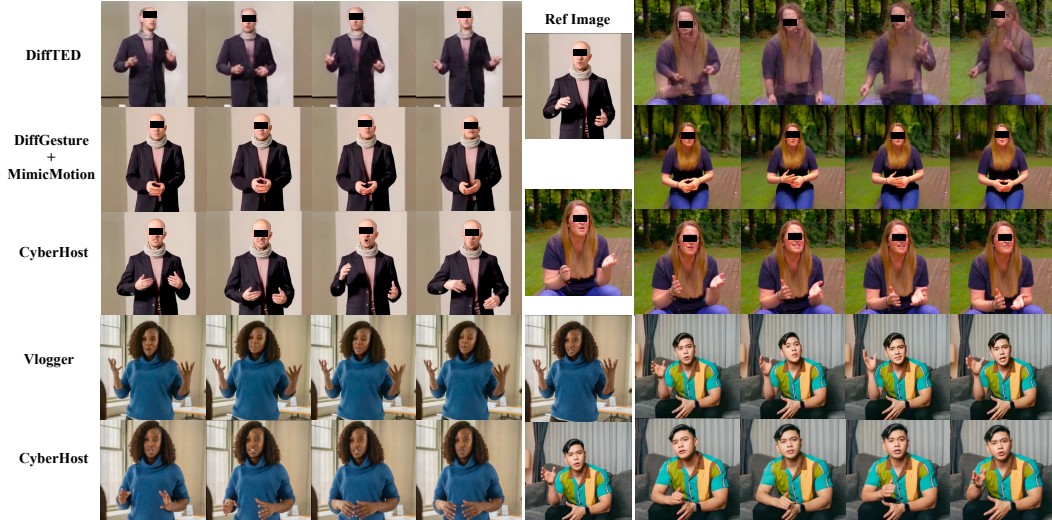

Figure 4: The audio-driven taking body results of CyberHost compared to other methods.

(CSIM) between the facial features of the reference image and the generated video frames. We use SyncC and SyncD, as proposed in (Prajwal et al., 2020), to evaluate the synchronization quality between lip movements and audio signals in audio-driven settings. Additionally, Average Keypoint Distance (AKD) is used to measure the accuracy of actions in video-driven settings. Because the AKD cannot be used to evaluate hand quality in audio-driven scenarios, we compute the average of Hand Keypoint Confidence (HKC) as a reference metric for evaluating hand quality. Similarly, we calculated the standard deviation of hand keypoint coordinates within a video segment as the Hand Keypoint Variance (HKV) metric to represent the richness of hand movements.

**Audio-driven Talking Body** Currently, only a few works such as Dr2 (Wang et al., 2023c), DiffTED (Hogue et al., 2024), and Vlogger (Corona et al., 2024) have adopted two-stage approaches to achieve audio-driven talking body video generation. However, these methods are not open-sourced, making it difficult to conduct direct comparisons. To better compare the effectiveness with the dual-stage method, we constructed a dual-stage audio-driven talking body baseline based on the current state-of-the-art audio2gesture and pose2video algorithms. Specifically, we trained DiffGesture (Zhu et al., 2023a) on our dataset to generate subsequent driving SMPLX (Pavlakos et al., 2019) pose sequences based on input audio and an initial SMPLX pose. Finally, the SMPLX meshes were converted into DWPose (Yang et al., 2023) key points, and MimicMotion (Zhang et al., 2024) was used for video rendering based on these key points.

Table 2: Quantitative comparison with existing video-driven body reenactment methods.

| Methods | SSIM↑ | PSNR↑ | FID↓ | FVD↓ | CSIM↑ | AKD↓ |
|---|---|---|---|---|---|---|
| DisCo (Wang et al., 2024b) | 0.660 | 17.33 | 57.12 | 1490.4 | 0.227 | 9.313 |
| AnimateAnyone (Hu et al., 2024) | 0.737 | 20.52 | 26.87 | 834.6 | 0.347 | 5.747 |
| MimicMotion (Zhang et al., 2024) | 0.684 | 17.96 | 23.43 | 420.6 | 0.340 | 8.536 |
| CyberHost (V2V-B) | **0.782** | **21.31** | **20.04** | **181.6** | **0.458** | **3.123** |

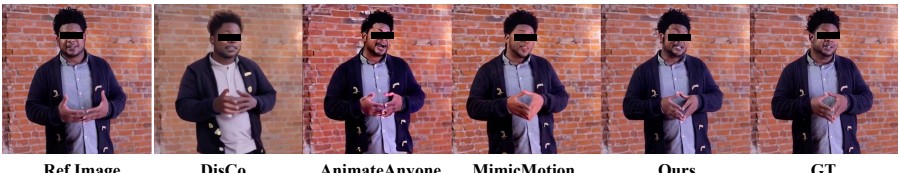

**Ref Image**     **DisCo**     **AnimateAnyone**     **MimicMotion**     **Ours**     **GT**

Figure 5: Comparisons with other video-driven body reenactment results

As shown in Table 1, our proposed CyberHost significantly outperforms DiffTED and the two-stage baseline in terms of image quality, video quality, facial consistency, and lip-sync accuracy in the audio-driven talking body (A2V-B) setting. It is important to note that due to DiffTED generating too many degraded hand results, the HKV metric can no longer reflect the diversity of hand movements. Figure 4 also presents a visual comparison between CyberHost and the two-stage baseline. Additionally, we utilized reference images and audio from 30 demos displayed on the Vlogger homepage to conduct both quantitative and qualitative comparisons with Vlogger. Notably, since most of Vlogger's test videos exhibit minimal motion, the HKC indicator is relatively high, whereas the HKV indicator, which measures the diversity of movements, is very low, as shown in Table 1. As depicted in Figure 4, our proposed CyberHost surpasses Vlogger in both generated image quality and the naturalness of hand movements. These results demonstrate that our one-stage framework, CyberHost, achieves significant improvements over existing methods across multiple key dimensions of synthesis quality, as further validated by the videos provided in the supplementary materials.

**Video-driven Body Reenactment**   We adapt our method to perform video-driven human body reenactment (V2V-B) by utilizing DWPose (Yang et al., 2023) to extract full-body keypoints from videos and replace the body movement maps with a sequence of skeleton maps. As shown in Table 2, we compared our CyberHost with several state-of-the-art zero-shot human body reenactment methods, including DisCo (Wang et al., 2024b), AnimateAnyone (Hu et al., 2024) and MimicMotion (Zhang et al., 2024). CyberHost significantly outperforms the current state-of-the-art methods in various metrics such as FID, FVD, and AKD. The visual results in Figure 5 also demonstrate that CyberHost achieves better structural integrity and identity consistency in local regions such as the hands and face compared to current state-of-the-art results.

**Audio-driven Talking Head.**   Although our framework is designed for talking body, it requires only minor modifications to be adapted for audio-driven talking head (A2V-H) setting. We removed the unnecessary Hand RAMs and adjusted the cropping area of the training data to focus around the face. As shown in Table 3, we compared our method with Hallo (Xu et al., 2024a), VExpress (Wang et al., 2024a) and EchoMimic (Chen et al., 2024b). We randomly sampled 100 videos from CelebV-HQ (Zhu et al., 2022) as the test set. Experimental results demonstrate that CyberHost surpasses current state-of-the-art performance across multiple metrics, including FID, FVD, and Sync score.

### 4.3 ABLATION STUDY

**Analysis of Region Attention Modules.**   As shown in Table 1, we conduct an ablation study to analyze the effectiveness of the Region Attention Modules. As we can see, the latents bank in RAM significantly improves metrics related to the hand quality like HKC and HKV. It can be observed that removing the temporal latents $L_{temp}$ from latents bank leads to a degradation in the hand motion modeling ability, as reflected by a decreased HKV, demonstrating the importance of spatial-temporal design. The identity descriptor, on the other hand, is more closely associated with the CSIM metric, demonstrating its effectiveness in maintaining identity consistency. Face RAM significantly improves facial-related metrics such as CSIM and Sync score, while Hand RAM effectively reduces

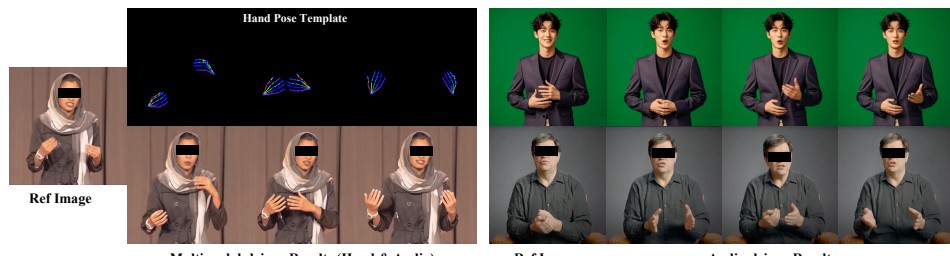

Figure 6: (a) Multimodal-driven results (b) Audio-driven results on the open-set test images.

Table 3: Comparison with existing audio-driven talking head methods.

| Methods | SSIM↑ | PSNR↑ | FID↓ | FVD↓ | CSIM↑ | SyncC↑ | SyncD↓ |
|---|---|---|---|---|---|---|---|
| EchoMimic (Chen et al., 2024b) | 0.619 | 17.468 | 35.37 | 828.9 | 0.411 | 3.136 | 10.378 |
| VExpress (Wang et al., 2024a) | 0.422 | 10.227 | 65.09 | 1356.5 | 0.573 | 3.547 | 9.415 |
| Hallo (Xu et al., 2024a) | 0.632 | 18.556 | 35.96 | 742.9 | 0.619 | 4.130 | 9.079 |
| CyberHost (A2V-H) | **0.694** | **19.247** | **25.79** | **552.6** | 0.581 | **4.243** | **8.658** |

artifacts in hand generation, thereby improving pixel quality metrics like FVD and FID. Local enhancement supervision enhances facial consistency, lip synchronization, and hand quality.

**Analysis of Human-Prior-Guided Conditions.** We also validated the effectiveness of various human-prior-guided conditions in Table 1. In the absence of body movement constraints, the motion space of the human root is unrestricted, resulting in unstable overall generation quality. This instability negatively impacts metrics related to overall generation quality, such as SSIM, PSNR, FID, and FVD, while the richness of hand movement metrics, that is HKV, may appear artificially high. Similarly, the absence of the hand clarity score as a prior for hand motion patterns results in more erratic hand movement trajectories. Frequent artifacts contribute to artificially high HKV, and significantly impact the hand quality metric, such as HKC. The pose-aligned reference features, by leveraging the topological structure priors of the reference images, also enhance the stability of the generated results, providing significant benefits in metrics such as HKC, FID, and FVD.

### 4.4 Multimodal-driven Video Generation

Our proposed CyberHost also supports combined control signals from multiple modalities, such as 2D hand keypoints and audio. As shown in Figure 6 (a), the hand keypoints are used to control hand movements and the audio signals are used to drive head movements, facial expressions, and lip synchronization. This driving setup leverages the explicit structural information provided by hand pose templates to enhance the stability of hand generation, while significantly improving the correlation of head movements, facial expressions, and lip synchronization with the audio.

### 4.5 Audio-driven Results in Open-set Domain

To validate the robustness of CyberHost, we tested the audio-driven talking body video generation results on open-set test images sourced from the internet and AIGC image synthesis. As shown in Figure 6 (b), our proposed method demonstrates good generalization across various characters and is capable of generating complex gestures, such as hand interactions.

## 5 Conclusion

This paper introduces a one-stage audio-driven talking body generation framework, CyberHost, designed to produce human videos that match the input audio with high expressiveness and realism. CyberHost features region attention module to enhance the details generation ability of key local regions and a suite of human-prior-guided conditions to reduce motion uncertainty in the audio-driven setting. Extensive experiments demonstrate that CyberHost can generate stable, natural, and realistic talking body videos and achieve zero-shot human image animation in open-set domains. Additionally, CyberHost can be extended to various human animation settings, achieving satisfactory results.

## 6 ACKNOWLEDGMENTS

We extend our gratitude to Dr. Pengfei Wei for assisting in setting up the two-stage audio-driven talking body video generation baseline.

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

## A    MORE EXPERIMENT RESULTS

### A.1    ADDITIONAL VISUALIZATION COMPARISONS

We present supplementary visual comparison results with state-of-the-art methods in the video-driven context, as illustrated in Figure 7. Additionally, we provide further visual results for open-set test images in the audio-driven context in Figure 8.

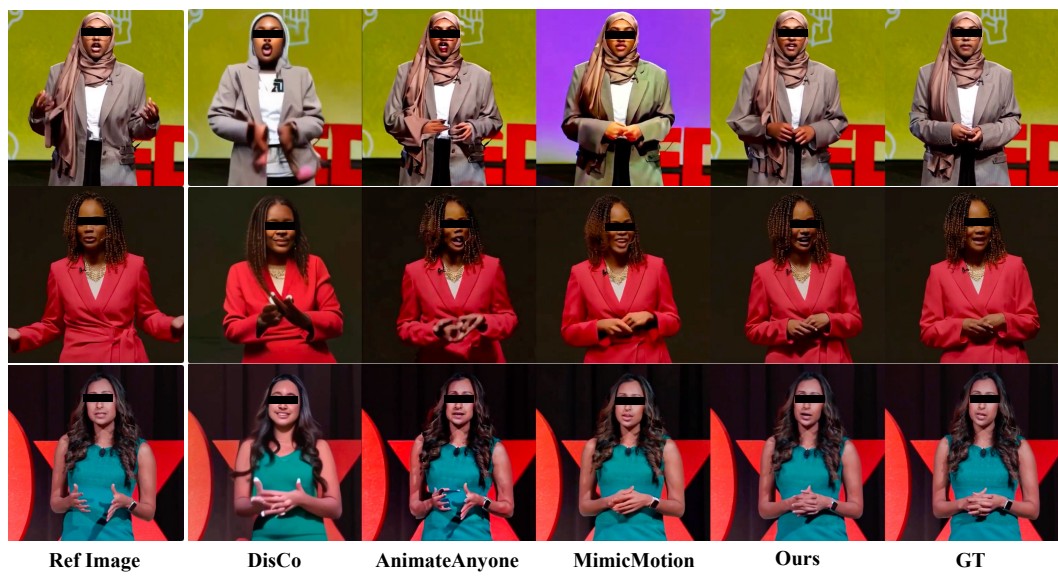

Figure 7: Additional comparisons results with other video-driven body reenactment results

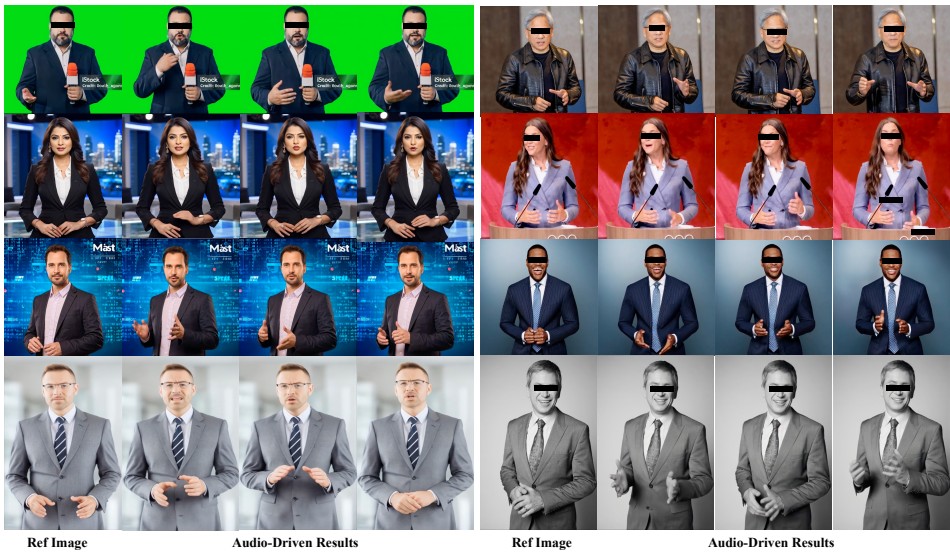

Figure 8: Additional audio-driven taking body results of CyberHost on the open-set test images.

### A.2    USER STUDY FOR SUBJECTIVE EVALUATION

We conducted a user study with five independent, professional evaluators. Specifically, we used the MOS (Mean Opinion Score) rating method. The specific evaluation criteria are discussed in

Table 4: User study of subjective evaluation. * denotes evaluate on vlogger test set.

| Method | ID Consistency↑ | Motion Naturalness↑ | Video Quality↑ | Lip-sync↑ |
|---|---|---|---|---|
| DiffTED | 1.54 | 1.52 | 1.92 | 1.00 |
| DiffGest.+MimicMo. | 3.28 | 2.56 | 3.32 | 2.34 |
| CyberHost (A2V-B) | 4.40 | 4.28 | 4.10 | 4.64 |
| GroundTruth | 4.94 | 4.72 | 4.66 | 4.88 |
| Vlogger * | 4.10 | 3.54 | 3.26 | 2.40 |
| CyberHost (A2V-B) * | 4.62 | 4.12 | 4.52 | 4.70 |

Section G. As can be seen from Table 4, Cyberhost shows performance advantages across multiple dimensions compared to the baseline methods.

### A.3 Ablation Study of Region Attention Module

To more effectively validate the effectiveness of our proposed Region Attention Module structure design, we have supplemented additional ablation experiments. These include removing the spatial latents, replacing the spatial latents and temporal latents with a unified 3D latents bank. As shown in Table 5, eliminating the spatial bank significantly affects the quality of the generated videos, while the 3D latents bank, with a 55-fold increase in parameters under the current configuration, did not provide further performance gain. We hypothesize that the spatial and temporal latents bank, which more closely aligns with the behavior of U-Net by decoupling the temporal attention and spatial attention, consequently leads to more efficient learning of local features. Additionally, as shown in the Figure 9, we conducted ablation experiments on different sizes of the latents bank in the RAM. The results indicate that increasing the size of the latents bank beyond the current configuration provides only marginal performance gains.

Table 5: More ablation results on Region Attention Module.

| Method | FID↓ | FVD↓ | SyncC↑ | HKC↑ | HKV↑ |
|---|---|---|---|---|---|
| w/o Regional Latents Bank | 37.80 | 643.9 | 6.384 | 0.859 | 21.35 |
| w/o Spatial Latents | 36.23 | 612.7 | 6.362 | 0.864 | 22.15 |
| w/o Temp. Latents | 36.75 | 624.1 | 6.468 | 0.870 | 19.44 |
| 3D Latents Bank | 32.94 | 552.3 | 6.613 | 0.888 | 23.93 |
| Spatial Latents + Temp. Latents | 32.97 | 555.8 | 6.627 | 0.884 | 24.73 |

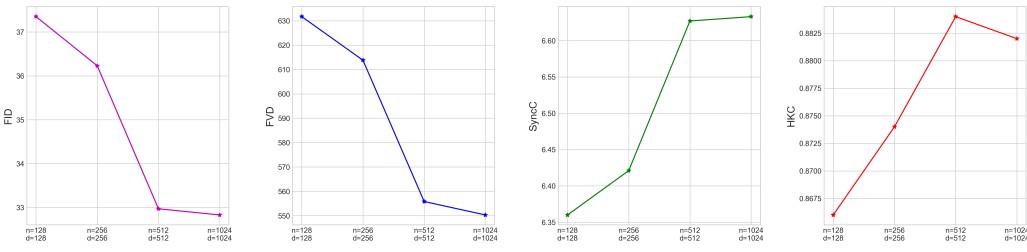

Figure 9: Comparison of results with different spatial latents sizes.

### A.4 Analysis of Regional Attention Masks from Different Layers

In Section 3.2, we employ auxiliary convolutional layers as a regional mask predictor to directly estimate a regional attention mask $M_{pred}$ to guide the optimization of the RAM modules. Note that we learn such a mask predictor in each block of the denoising U-Net. Since different layers learn different kinds of features to serve different roles in the network, we also investigated the accuracy

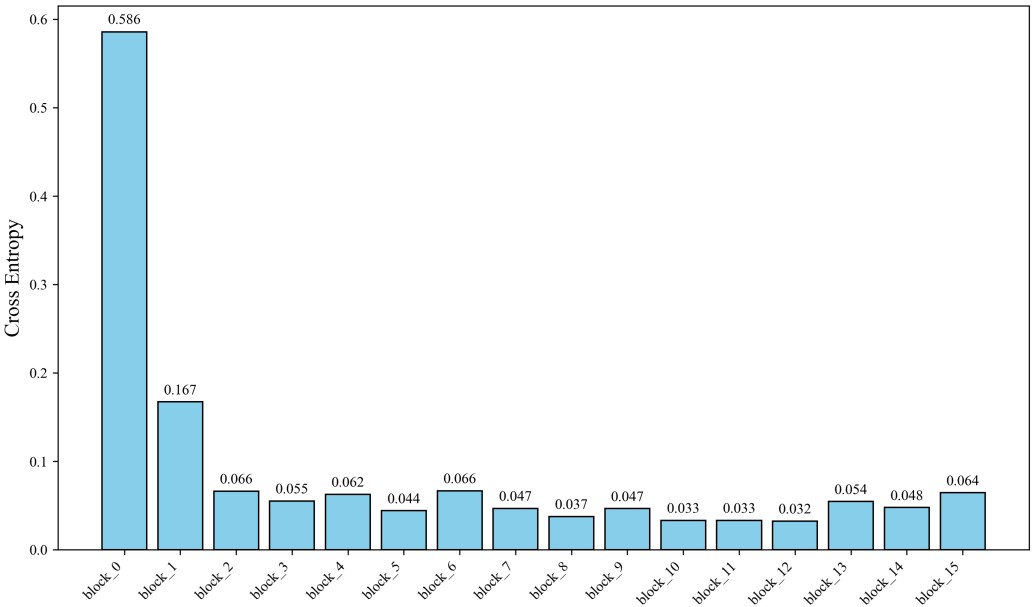

Figure 10: The accuracy of hand attention mask predictions across different layers in the denoising U-Net.

Table 6: Quantitative comparison of inserting RAMs at different layers.

| Method | FID↓ | FVD↓ | CSIM↑ | SyncC↑ | HKC↑ |
|--------|------|------|-------|--------|------|
| top-1 | 39.22 | 664.3 | 0.421 | 6.271 | 0.853 |
| top-5 | 36.44 | 594.0 | 0.470 | 6.512 | 0.869 |
| top-10 | 33.88 | 573.1 | 0.489 | 6.549 | 0.874 |
| All | 32.97 | 555.8 | 0.514 | 6.627 | 0.884 |

differences in hand attention mask predictions across different layers. As shown in Figure 10, we calculated the cross-entropy between the predicted hand mask from each layer and the hand region in the generated results on a batch of test data to measure the effectiveness of each attention block. Based on the accuracy of the predicted masks from each layer, we created corresponding top-1, top-5, and top-10 experiments (where "top" refers to lower cross-entropy) to verify whether predicting masks and inserting RAM module in fewer layers could yield better results. As shown in Figure 6, We found that using RAM modules in more simplified layers was not as effective as using them in all layers, which may suggest that even layers with less accurate hand mask predictions still have the necessity to enhance local features learning.

## A.5 ABLATION STUDY OF HAND CLARITY SCORE

As shown in Figure 11, we show the impact on the generation results when different hand clarity score conditions are input. It can be seen that it will directly affect the generation quality and dynamic degree of gestures. As illustrated in Figure 13, the hand clarity score reflects the clarity of the hands in the training data, thereby allowing the identification of training videos with varying quality of hand visuals. It enables the model to distinguish between high-quality and low-quality hand images during training, thereby guiding the model to generate clear hand visuals during inference by inputting a high hand clarity score.

## A.6 COMPARISON WITH ANGIE

ANGIE (Liu et al., 2022) establishes a unified framework to generate speaker image sequences driven by speech audio. However, it primarily focuses on designing a vector quantized motion

extractor to summarize common co-speech gesture patterns, and a co-speech gesture GPT with motion refinement to complement the subtle prosodic motion details. The overall emphasis of this work is on the audio-to-motion generation aspect, utilizing a pre-existing image generator model from MRAA (Siarohin et al., 2021) for the visual synthesis. While our proposed CyberHost aims to optimize both motion generation and image generation simultaneously, providing a more unified algorithmic framework. In Figure 12, we provide a comparison with ANGIE. It is important to note that ANGIE is trained and tested simultaneously on the four individuals in the PATS (Ahuja et al., 2020) dataset, whereas CyberHost focuses on the driving performance for open-set images. Despite not specifically training on the PATS dataset, our generated results exhibit natural motion patterns and significantly better video clarity. Additionally, we performed a visual comparison on the Speech2Gesture(Ginosar et al., 2019) dataset against methods such as Speech2Gesture (Ginosar et al., 2019), MoGlow (Alexanderson et al., 2020), and SDT (Qian et al., 2021). Please refer to the supplementary materials for the comparison video.

# B  ADDITIONAL DATASET DETAILS

## B.1  DATASET COLLECTION

Most of our data primarily originates from open-source datasets (Lin et al., 2024; Chen et al., 2024a), as well as from open platforms that adhere to the "Creative Commons" license, such as Pexels and Pixabay. All collected videos are publicly accessible, and we employ professionals to review the data sources to ensure compliance with proper licensing agreements. Our emphasis is on half-body human scenes; thus, we filter out data unrelated to human subjects based on video titles and other metadata. Subsequently, through a series of data processing steps, we curated a collection of single-person, half-body speaking videos, ensuring that key regions, such as the hands and head, remained within the frame. All collected video data is converted into latent embeddings using a Variational Autoencoder (VAE) encoder and identified via MD5 checksums. This approach eliminates the storage of the videos' RGB content and URLs, thereby safeguarding any personal information related to the subjects. Additionally, we have implemented strict access controls to prevent non-research personnel from accessing the data. Next, we will provide a detailed explanation of several key steps in our data processing pipeline.

1. **Video Trimming**: We utilized PySceneDetect (https://github.com/Breakthrough/PySceneDetect) to trim shot transitions and fades in video clips. The trimming method employed is akin to SVD (Blattmann et al., 2023a), but we adjusted the parameters specifically for human body scenes.

2. **Half-body Cropping**: Using open-source tools like DWPose (Yang et al., 2023), we detected human key points and cropped the body region. We excluded video clips with more than two individuals, those where the subject was too small within the frame, and those with excessive human movement. This filtering allowed us to select half-body videos that met our criteria based on key points.

3. **Exclusion of Background Motion**: We applied foreground segmentation algorithms to identify the background regions within frames. We calculated the per-pixel differences between frames in the background areas and excluded data with significant background movement.

4. **Audio-Video Synchronization**: In the final stage of data processing, we used tools like LipSync to verify the synchronization between video and audio lip movements, discarding any video data with incorrect audio.

Table 7: Statistics of dataset scale of different methods.

| Method | Clips No. | Duration | IDs No. |
|---|---|---|---|
| DiffTED (Hogue et al., 2024) | 1134 | 2.75h | 353 |
| Vlogger (Corona et al., 2024) | - | 2200h | 800k |
| CyberHost | 104k | 200h | 10k |

Table 8: Data Distribution in the Dataset.

| Category | Subcategory | Proportion |
|---|---|---|
| Gender | Male | 63.1% |
| | Female | 36.9% |
| Environment | Indoor | 85.2% |
| | Outdoor | 14.8% |
| Video Duration | $< 5s$ | 58.2% |
| | 5s-10s | 41.8% |
| Resolution Area | $384^2$ - $512^2$ | 45.3% |
| | $> 512^2$ | 54.7% |

### B.2 DATASET STATISTICS

Table 7 presents statistics on the number of clips, total duration, and number of IDs in our training data, comparing these with existing audio-driven talking body generation works (Hogue et al., 2024; Corona et al., 2024). DiffTED (Hogue et al., 2024), a GAN-based method, shows lower overall generation quality due to its lesser reliance on data. Both Vlogger (Corona et al., 2024) and our proposed CyberHost employ diffusion-based methods; however, despite having significantly less training data duration and fewer IDs compared to Vlogger, CyberHost achieves noticeably higher generation performance. This suggests that our performance improvements are derived from the efficacy of our algorithmic design rather than the quantity of data used. Additionally, we analyze the dataset's distribution across four dimensions: gender, environment, video duration, and resolution area, as shown in Table 8.

## C ETHICS IMPACTS

Cyberhost enables the creation of realistic talking body videos using audio and a single image. While this technological advancement offers significant potential for fields such as education and commercial advertising, greatly enhancing and enriching people's lives, it also raises ethical concerns. Caution is required to prevent its misuse in generating offensive or potentially infringing deepfake videos. We have carefully considered various strategies to regulate video generation within the Cyberhost application. 1) We need to confine the usage scenarios of this technology, authorizing its application solely in legitimate areas such as education and commercial advertising. This restriction aims to prevent the creation of illicit or infringing video content. Additionally, embedding the creator's digital fingerprint into the video will facilitate legal tracing. 2) We plan to embed visible watermarks in the generated videos to help individuals identify AI-generated content, thereby preventing potential misguidance. 3) We aim to establish a thorough review system to ensure the safety of user-uploaded images and audio within Cyberhost. For images, techniques such as pornography detection and celebrity recognition can be employed. For audio, Automatic Speech Recognition (ASR) can transcribe audio to text, allowing for the analysis of potential issues using Large Language Models. This approach helps prevent the generation of videos containing offensive audio.

## D CALCULATION OF HAND CLARITY SCORE

We devised the hand clarity score to mitigate the impact of hand blur data on the learning of hand structures within the dataset. Consequently, this metric aims to quantify the degree of blurriness in the hand regions of the images. This is achieved by assessing the detail richness and edge sharpness of the hand images. We use DWPose (Yang et al., 2023) to extract body keypoints and form bounding boxes for the left and right hands. As illustrated in Figure 13, note that the areas corresponding to the left and right hands may overlap. Subsequently, the hand images are resized to a fixed resolution $128 \times 128$ to ensure the statistical analysis remains meaningful.

Given a video clip with a frame length of $n$, as detailed in Algorithm 1, we apply a Laplacian transformation to hand images and calculate the standard deviation, denoted as $L_{1:n}^{left}$ and $L_{1:n}^{right}$. This metric indicates the clarity of the hand images, with higher values representing less blurriness. Since

the absolute standard deviation of the Laplacian can vary in scale depending on the image subject, we also introduce the relative standard deviation of the Laplacian, denoted as $\hat{L}_{1:n}^{left}$ and $\hat{L}_{1:n}^{right}$. Specifically, we select the frame corresponding to the 90th percentile of $L_t^{left}$ within this video clip as the pivot frame, which is treated as a clear-hand frame. Both the absolute and relative Laplacian standard deviations are considered hand clarity scores and are subsequently fed into denoising U-Net as a condition, enabling the model to gauge the appropriate level of blurriness for the output image. During the inference stage, a high clarity score is input into CyberHost, prompting the model to generate stable and clear hand videos.

---

**Algorithm 1** Hand Clarity Score Extraction

---

**INPUT:** Original video frames $I_{1:n}^{ori}$

  Extract human keypoints $K_{1:n}^{ori}$ from $I_{1:n}^{ori}$

  Crop out left & right hand areas in terms of $K_{1:n}^{ori}$ from $I_{1:n}^{ori}$ and resize them to $128 \times 128$ resolution, denoted as $I_{1:n}^{left}$ and $I_{1:n}^{right}$

  Calculate relative Laplacian standard variance with $KernelSize = 3$ from $I_{1:n}^{left}$ and $I_{1:n}^{right}$, denoted as $L_{1:n}^{left}$ and $L_{1:n}^{right}$

  Assign $L_{pivot}^{left}$ and $L_{pivot}^{right}$ to be the 90% percentile value of $L_{1:n}^{left}$ and $L_{1:n}^{right}$

  Calculate absolute Laplacian standard variance $\hat{L}_{1:n}^{left} \leftarrow L_{1:n}^{left}/L_{pivot}^{left}$, and the same for $\hat{L}_{1:n}^{right}$

  **return** $L_{1:n}^{left}$, $L_{1:n}^{right}$, $\hat{L}_{1:n}^{left}$ and $\hat{L}_{1:n}^{right}$

---

## E  ADDITIONAL IMPLEMENTATION DETAILS

In the region attention module, we set $n$, $m$, and $d$ to 512, 64, and 512, respectively. Consequently, this configuration results in a spatial latents $L_{\text{spa}}$ of shape $(1, 512, 512)$, and a temporal latent $L_{\text{temp}}$ of shape $(1, 64, 512)$. Both $L_{\text{spa}}$ and $L_{\text{temp}}$ are initialized using Gaussian noise, with the Gram-Schmidt process being applied to both during each forward pass to ensure orthogonality. ROPE embedding is utilized to inject temporal and spatial positional information into the latents bank, respectively. As for the regional attention mask $\mathbf{M}_{\text{pred}}$ in Section 3.2, we employ MSE loss to supervise the learning process. The hand mask loss and face mask loss are weighted by 0.005 and 0.001, respectively, in the total loss. For the input audio feature, to fully leverage multi-scale audio features, we concatenate the hidden states from each encoder layer of the Wav2Vec (Schneider et al., 2019) model with the last hidden state, resulting in an audio feature vector of size 10752. To capture more temporal context, we use a temporal window of size 5, constructing an input audio feature of shape $(5, 10752)$ for each video frame. It is important to note that both the video-driven body reenactment model and the multimodal-driven video generation model require separate training processes. During the training process for video-driven body reenactment, we replace the body movement map with the full-body skeleton map. For multimodal-driven video generation, we directly draw the skeleton maps of both hands onto the body movement map.

The training process of our audio-driven talking body generation model is divided into two stages. In stage one, The training was conducted for a total of 4 days on 8 A100 GPUs, with a batch size of 12 per GPU. In the second stage, We use a total of 32 A100 GPUs to train for 4 days, with each GPU processing only one video sample. This setup allows us to train with different resolutions on different GPUs, enabling convenient dynamic resolution training. We constrain these different resolutions to have an area similar to the $640 \times 384$ resolution, with both the height and width being multiples of 64 to ensure compatibility with the LDM structure. During inference, we use a dual classifier-free guidance (CFG) strategy for both the reference image and audio, with scales set to 2.5 and 4.5, respectively. Specific implementation can be referenced in (Karras et al., 2023). We found that a higher reference image CFG scale leads to oversaturated images, while a higher audio CFG scale results in unstable movements. The model achieves an overall inference Real-Time Factor (RTF) of approximately 65 on a single A100 GPU.

## F  LIMITATIONS AND FUTURE WORK

We will explore the current limitations and future work in four distinct directions.

**More Realistic.** While our proposed CyberHost model aims to mitigate issues of detail underfitting and motion uncertainty in one-stage audio-driven talking body generation, there are still instances of localized degradation in certain body regions, such as hands and face. Additionally, our generated results exhibit noticeable artifacts, particularly in hand details, and the overall performance has yet to reach the Turing test level achieved by methods like EMO (Tian et al., 2024), which are specialized in talking head generation. As such, enhancing the quality of key area details and improving the realism of character movement patterns represent critical areas for future work.

**More Robust.** CyberHost continues to face challenges in generating high-quality videos from certain difficult inputs. For instance, as illustrated in Figure 14, exaggerated body proportions in cartoon images often result in unrealistic outputs. Conversely, the driving effect is more effective for cartoon images with body proportions closer to those of real humans. Moreover, complex backgrounds resembling hand colors, such as skin-toned backgrounds, are prone to generating errors. Challenging human poses, such as raising both hands above the head, further complicate the generation of natural motion. Regarding the audio inputs, scenarios with complex background noises and varying tones, such as singing, pose additional challenges for the model. These conditions make it difficult to generate accurate lip movements and naturally synchronized body movements. Examples of these challenging scenarios can be found in the videos provided in the supplementary material. For these challenging test images, our future work involves collecting a more diverse dataset of character videos, including anime scenes and data featuring complex human poses. Additionally, it will be necessary to design a more effective architecture that encodes prior knowledge of human body structures into the diffusion model, thereby improving its understanding of the subjects in diverse test images. Regarding complex test audio, we aim to replace Wav2Vec (Schneider et al., 2019) with a more advanced audio feature extractor. Furthermore, implementing background noise reduction strategies will enhance the model's robustness to audio variations.

**Full-Body Generation.** On the other hand, despite the fact that our training data does not include videos of full-body scenes, we still present the audio-driven results on full-body test images in Figure 14. It can be observed that directly testing our current model on full-body scenes leads to significant degradation of generated details in areas such as the face and hands. Additionally, the overall body proportions and movements appear quite unnatural. The primary reasons for the suboptimal performance of the current model on full-body scenarios are twofold. Firstly, our training data primarily consists of upper-body scenes, which limits the model's ability to generalize effectively to full-body scenarios, resulting in errors and unnatural limb generation. Secondly, at the current generation resolution ($640 \times 384$), the face and hand regions in full-body videos occupy a smaller proportion of the frame. The existing VAE and U-Net architectures have limited capability to model faces and hands effectively at such a low resolution. In future work, we are keen to collect data that includes full-body scenarios and to increase the model's resolution to better support video generation in full-body scenes.

## G  MEAN OPINION SCORE CRITERIA

**Identity Consistency:**

- **1:** Identity is not preserved at all; the generated identity does not resemble the original in any aspect.
- **2:** Identity is faintly preserved; the output shows some resemblance to the original, but major inconsistencies are apparent.
- **3:** Identity is moderately preserved; while the general likeness is maintained, there are notable discrepancies.
- **4:** Identity is well-preserved; only minor differences are noticeable.
- **5:** Identity is perfectly preserved; the output matches the input without any discernible inconsistencies.

**Motion Naturalness:**

- **1:** Motion appears highly unnatural; movements are jerky or disjointed.
- **2:** Motion is somewhat unnatural; movements have notable unnatural transitions.

- **3:** Motion has a mix of natural and unnatural elements; movements are generally smooth with some unnatural aspects.
- **4:** Motion appears mostly natural; only minor unnatural movements are present.
- **5:** Motion is completely natural; movements are fluid and appear realistic.

**Video Quality:**

- **1:** Video quality is very poor; heavy artifacts, blurring, and pixelation are present.
- **2:** Video quality is below average; significant artifacts and blurring occur frequently.
- **3:** Video quality is average; some artifacts and blurring, but generally acceptable.
- **4:** Video quality is good; few artifacts and blurring instances are minimal.
- **5:** Video quality is excellent; no noticeable artifacts or blurring.

**Lip-Sync Accuracy:**

- **1:** Lip movements are completely out of sync with the audio.
- **2:** Lip movements are mostly out of sync; frequent discrepancies between audio and video.
- **3:** Lip movements are partially in sync; occasional mismatches between audio and video.
- **4:** Lip movements are generally in sync; minor mismatches are observed.
- **5:** Lip movements are perfectly in sync with the audio.

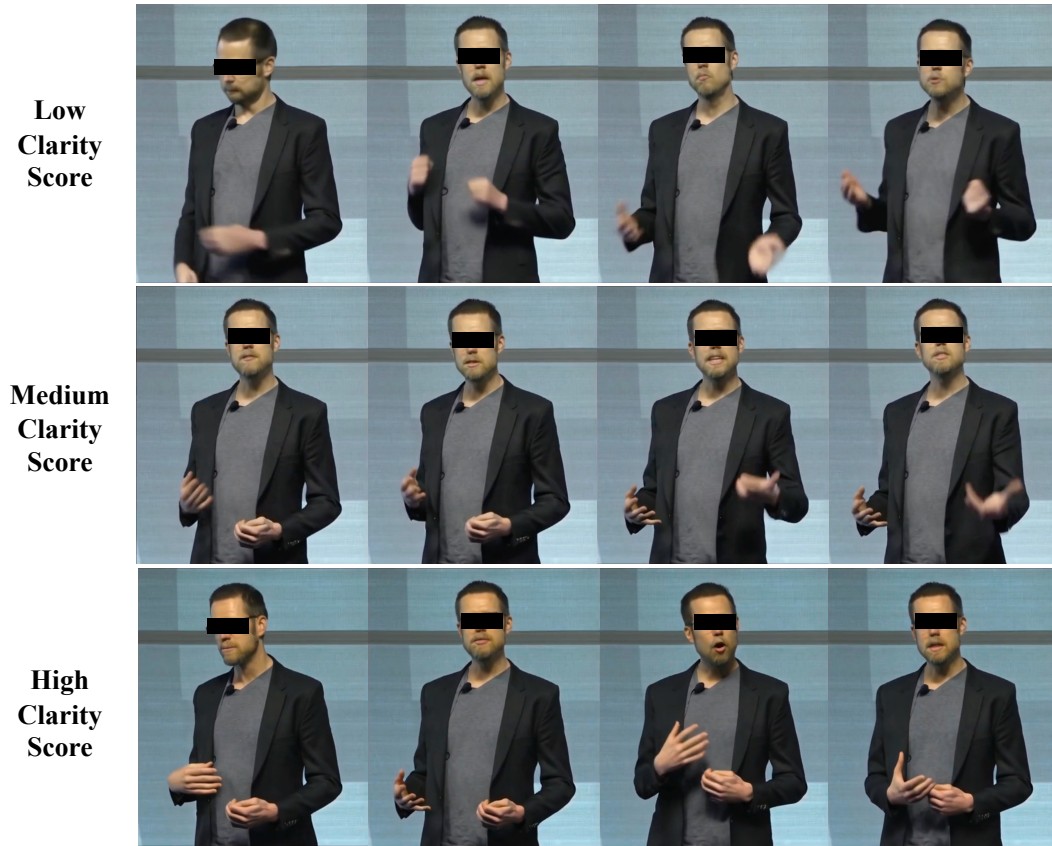

Figure 11: Visualization of videos generated with different hand ablation scores.

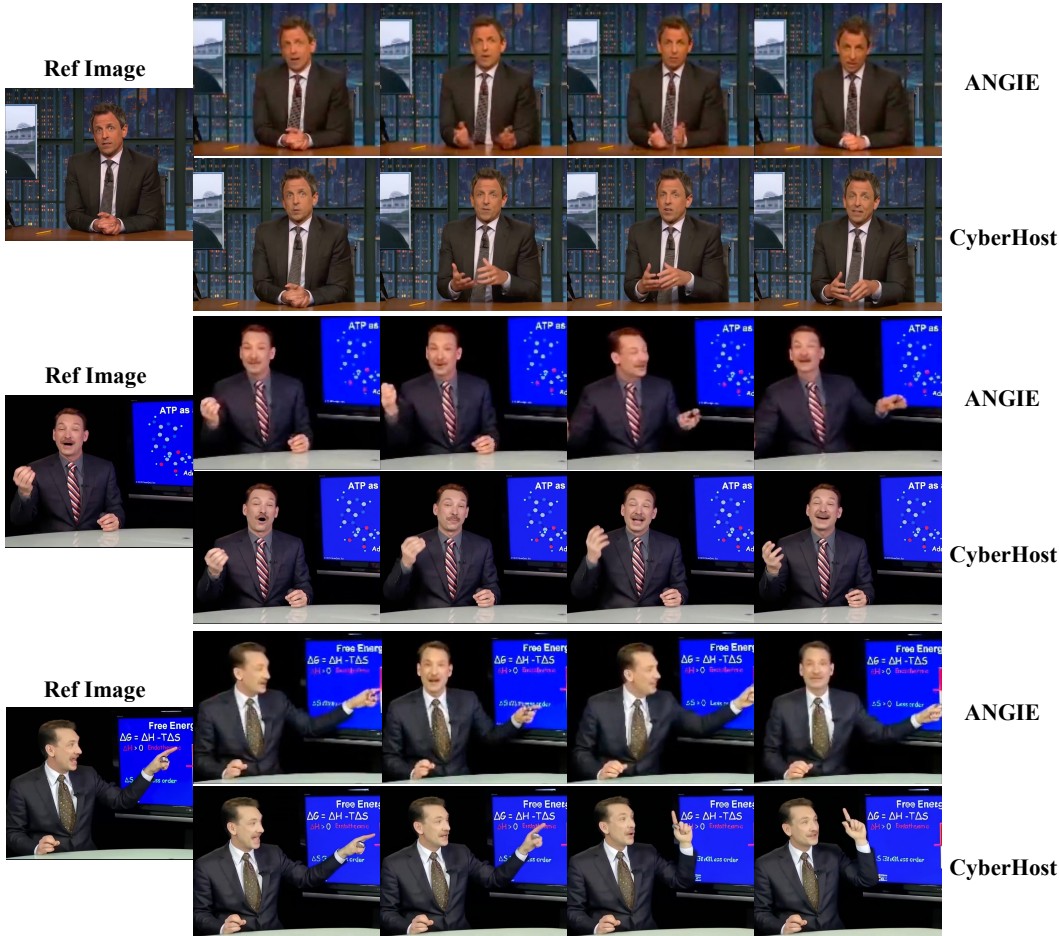

Figure 12: Comparison with ANGIE in the PATS dataset.

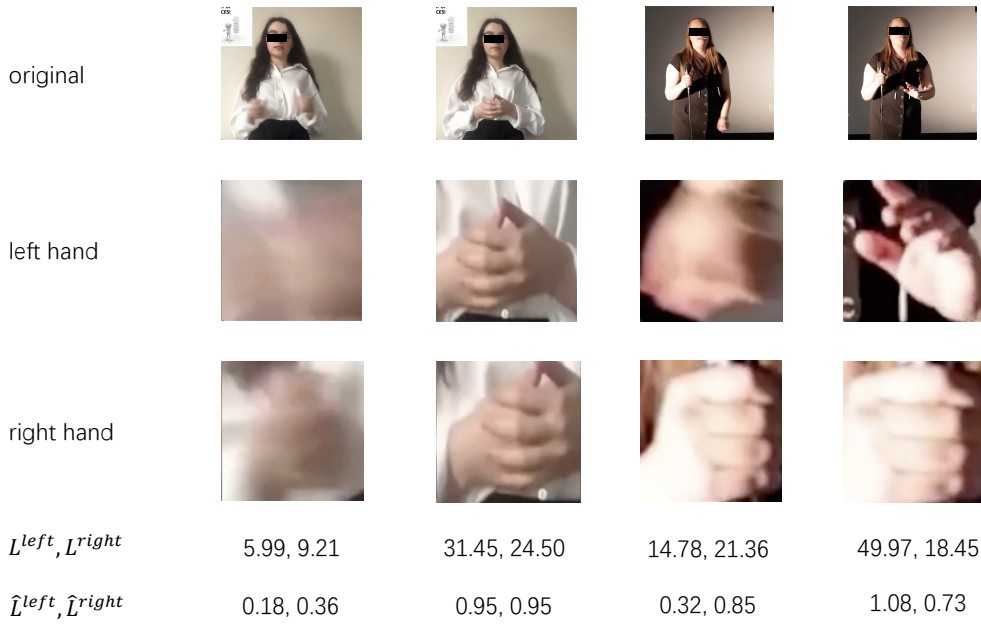

| | | | | |
|---|---|---|---|---|
| original | | | | |
| left hand | | | | |
| right hand | | | | |
| $L^{left}, L^{right}$ | 5.99, 9.21 | 31.45, 24.50 | 14.78, 21.36 | 49.97, 18.45 |
| $\hat{L}^{left}, \hat{L}^{right}$ | 0.18, 0.36 | 0.95, 0.95 | 0.32, 0.85 | 1.08, 0.73 |

Figure 13: Cases on cropped hand images and the calculated hand clarity score.

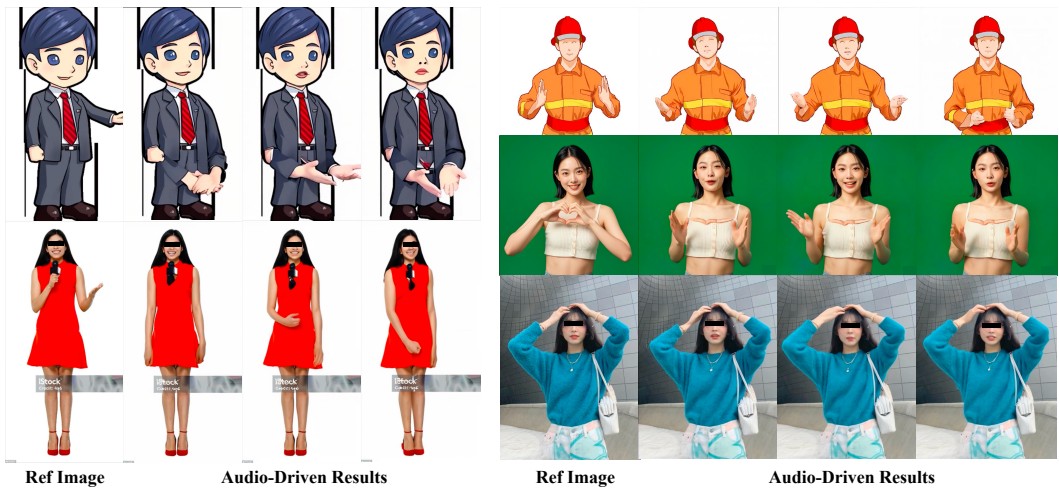

Ref Image      Audio-Driven Results      Ref Image      Audio-Driven Results

Figure 14: Failure cases of CyberHost in challenging scenarios.

