# OpenReview forum: "CyberHost: A One-stage Diffusion Framework for Audio-driven Talking Body Generation"
_ICLR.cc/2025/Conference — ICLR 2025 Oral_

### Official Review · Reviewer_FbGb · 2024-10-17

**Soundness:** 3
**Presentation:** 3
**Contribution:** 3
**Rating:** 8
**Confidence:** 3

**Summary:**

This paper introduces CyberHost, audio-driven human animation framework based on diffusion models. It addresses the less explored area of full-body human animation driven by audio signals, focusing on enhancing the generation quality of critical regions like the hands and face. The authors propose a Region Codebook Attention mechanism, along with a suite of human-prior-guided training strategies. The paper aims to bridge the gap in audio-driven human animation by improving hand clarity and overall natural motion.

**Strengths:**

The authors aimed to tackle two significant challenges in audio-driven body generation and achieved progress in:
1. Improving the synthesis quality of critical regions (hands and face)
2. Reducing motion uncertainty caused by weak correlations.

Specifically, this paper successfully addresses the challenge of generating high-quality hand and facial features using proposed modules including RAM.

In addition, comprehensive experiments were conducted. Comparisons were made to evaluate not only audio-to-body generation methods but also video-to-video and audio-to-face methods, demonstrating its expandability.

**Weaknesses:**

While most parts are understandable, some details and explanations are missing. The questions regarding the missing information are listed under "Questions." Additionally, as the methods utilize many well-known architectures and frameworks while introducing several modules—including the Latent Bank, Pose Encoder, Heatmap Estimator, and Mask Predictor—some missing information limits the paper’s reproducibility and clarity of the paper. If the concerns or questions listed on "Questions" are addressed, this paper would be worthy of a higher rating.

**Questions:**

1. In Section 3.2, how was the hand heatmap estimator trained? Was it trained jointly with Equation 6 during stage 1, stage 2, or was it pretrained former to Equation 6? Also, when training the hand heatmap estimator, were all weights shared across timesteps?

2. Are the Pose Encoder in the Body Movement Map and the Pose-Aligned Reference Feature shared? If they are, why are the rectangular box and human pose encoded using a shared network? What are the advantages of using a shared network compared to using different networks that share the latent space? If they are not shared, they should not be described as using the same Pose Encoder or abbreviated as "P."

3. Were the diffusion models initialized with pretrained weights or trained from scratch? At first, it seemed they were being trained from scratch, but in Line 191, it states, "we extend the 2D version to 3D by integrating the pretrained temporal module from AnimateDiff." Could you clarify how all the components were initialized?


Simpler Questions

4. What are the dimensions of L_spa and L_temp in Latent bank?

5. Starting from Line 855, how will this review system be incorporated into practical applications and future research?

6. Is Laplacian standard variance sufficient for "Hand clarity score?"

**Details Of Ethics Concerns:**

Due to the subject beign human video generation, careful and responsible appoarch is required.

---

> ### Author Response · Authors · 2024-11-22
> **Author Response to Reviewer FbGb**
>
> We are grateful for your positive review and valuable comments, and we hope our response fully resolves your concerns.
>
> >Q1. About the hand heatmap estimator
>
> Specifically, we added several convolutional layers in each hand RAM module to serve as the hand keypoint heatmap estimator, with weights shared across timesteps. The right part of Equation 6 represents the supervision signal for the hand heatmap estimator. Since we only train our RAM modules in stage 2, the hand heatmap estimator is also updated during stage 2 along with the loss in Equation 6.
>
> >Q2. About the Pose Encoder
>
> Thank your very much for your reminder, we apologize for any confusion caused by our unclear description. The Pose Encoder in the Body Movement Map and the one in the Pose-Aligned Reference Feature share the same model architecture, except for the first layer, but do not share any model parameters. Based on your suggestion, we have modified the corresponding symbols in Figure 2 and added clearer descriptions in the relevant paragraphs of the main text to distinguish between these two pose encoders.
>
> >Q3. About the model initialization
>
> We apologize for any confusion caused by our unclear description. The base denoising U-Net and reference net are both initialized with SD1.5 pretrained weights. We inserted the same temporal layers as AnimateDiff into the basic U-Net architecture to support sequential video prediction. To accelerate convergence on video data, the temporal attention layer weights are initialized with AnimateDiff's pretrained weights.
>
> >Q4. About the dimension of  Latent bank
>
> Thank you for your reminder. In the region attention module, we set **n**, **m**, and **d** to 512, 64, and 512, respectively. Consequently, this results in a spatial latent of shape (1, 512, 512), and a temporal latent of shape (1, 64, 512). We have clarified these settings in Section D of the revised appendix.
>
> >Q5. About the Review System
>
> Thank you for your interest in this section. Conducting safety reviews of generated content is crucial for the deployment of any AIGC task. For CyberHost, we are planing to develop a review system to ensure the safety of user-uploaded images and audio. For images, techniques such as pornography detection and celebrity recognition can be employed. For audio, Automatic Speech Recognition (ASR) can convert audio to text, which can then be analyzed for potential issues using Large Language Models, thereby preventing the video generation with offensive audio.
>
> >Q6. About the detection of hand clarity score
>
> (1) About the selection of detection methods, first of all, we have tried to use the IQA expert model [1, 2] and the MLLM model [3, 4] to detect the blurriness of the crop hand image, but found that the effects are not good. (2) In the [video](https://cyberhostanony.github.io/static/videos/rebuttal/HAND_CLARITY_ABLATION.mp4), we show the impact on the generation results when different hand clarity score conditions are input. It can be seen that it will directly affect the generation quality and dynamic degree of gestures. As illustrated in Figure 13 of the paper, the hand clarity score reflects the clarity of the hands in the training data, thereby allowing the identification of training videos with varying quality of hand visuals. It enables the model to distinguish between high-quality and low-quality hand images during training, thereby guiding the model to generate clear hand visuals during inference by inputting a high hand clarity score. This is also supported by the ablation experiment in Table 1.
>
> [1] Su S, Yan Q, Zhu Y, Zhang C, Ge Xin, Sun J, Zhang Y. Blindly Assess Image Quality in the Wild Guided by a Self-Adaptive Hyper Network
>
> [2] Wu H,  Zhang E, and Liao L, and et.al. Towards Explainable Video Quality Assessment: a Database and a Language-prompt Approach.
>
> [3] Wu H, Zhu  H,  Zhang Z and et.al. Towards Open-ended Visual Quality Comparison
>
> [4] Wu H, Zhang Z, Zhang W and et.al. Q-Align: Teaching LMMs for Visual Scoring via Discrete Text-Defined Levels
>
>
> **Summary**
>
> Following your comments, we added discussions on the hand heatmap estimator, pose encoder, model initialization, the dimension of the latent bank, review system, and hand clarity score in the revised appendix. We think this enhances the paper and better explains the method. Thank you again for your valuable comments and positive recommendation for our paper.

---

> ### Comment · Reviewer_FbGb · 2024-11-24
>
> Thank you for authors for the rebuttal.
>
> I have reviewed the comments, including those from other reviewers and responses. Most of the concerns I previously had have been addressed, which led me to raise my score. Furthermore, additional studies conducted during the discussion (rebuttal) period provided valuable insights.
>
> Related to Q1 hand heatmap estimation, one suggestion for future work would be to analyze the timesteps, as different timesteps may contain distinct feature information.

---

> > ### Author Response · Authors · 2024-11-24
> > **Author Response to Reviewer FbGb**
> >
> > Thank you for your valuable suggestions and positive feedback.
> >
> > Exploring the effectiveness of regional supervision at different timesteps is indeed a crucial optimization direction, and we will continue to delve into this aspect in our future work. Your insights are highly beneficial for our ongoing and future research efforts, and we sincerely appreciate your thoughtful recommendations.

---

### Official Review · Reviewer_1qwV · 2024-10-24

**Soundness:** 3
**Presentation:** 3
**Contribution:** 3
**Rating:** 10
**Confidence:** 5

**Summary:**

This paper proposes a novel and elegant one-stage audio-driven human diffusion model. The authors primarily focus on the most challenging problems of existing body animation models, which are details underfitting and motion uncertainty. To address details underfitting, the authors introduce a region attention module, and to tackle motion uncertainty, they design a series of human-prior-guided conditions. The paper is well-written and enjoyable to read. The final video results demonstrate high-quality rendering and natural motion driving.

**Strengths:**

1. This paper addresses two important and challenging problems in the body animation field, and the proposed approaches are novel and effective.
2. The proposed method supports multi-modalities driving
3. The driving results show really good rendering quality and natural motion fidelity.
4. The paper is well-organized and well-written.

**Weaknesses:**

1. Some details are not provided:
a) Which specific layers of Wav2Vec features were used? (line 191)
b) How to constrain the basis vectors of the latents bank to be orthogonal? (line 242)
c) There is a lack of loss description for regional mask predictor. (line 260)
2. Authors claim that the hand clarity score can enhance the model's robustness to blurry hands during training and enable control over the clarity of hand images during inference. They conducted ablations on hand clarity, but they did not demonstrate to what extent this score can control hand clarity during inference. I would like to know this result.
3. The explanation of how the proposed 'Pose-aligned Reference Feature' works has not convinced me for two reasons:
a) Although the ablation on pose-aligned ref shows a lower HKC score compared with Cyberhost, this method was proposed to solve the case of challenging initial poses, and the authors did not demonstrate its effectiveness in that scenario.
b) The authors claimed that the skeleton map provides topological structure information, which improves the quality of hand generation. However, they did not explain how this structural information actually contributes to generating higher-quality hand images.
4. Some spelling mistakes: 'feference' should be corrected to 'reference' in line 313.

**Questions:**

1. The authors claimed that the two-stage methods are mainly limited by the capability of the pose or mesh detectors, this limitation constrains the model's ability to capture subtle human nuances. I wonder if there exists, for example, a mesh detector that provides accurate and nuanced results. What are the advantages of a one-stage method compared to a two-stage method?
2. The authors presented various driving results, including video-driven body reenactment and multimodal-driven video generation. Was the model retrained when performing these two types of driving cases? If not, why can the body movement map be directly replaced by a skeleton or hand pose template?
3. Is the regional mask predictor embedded in all layers? Because different layers learn different kinds of features to serve different roles in the network. Therefore, I wonder about the effectiveness of predicting regional masks in all layers. Perhaps predicting the mask from the most effective layer could perform better.

Considering the good results and novelty, I would be very willing to raise my rating if my questions are answered.

New after rebuttal: The authors provide solid extended experiments, more implementation details, and reasonable explanations about the proposed questions. Taking the good results and novelty into consideration, I think this is really a good paper. Therefore, I decide to raise my rating to 10.

---

> ### Author Response · Authors · 2024-11-22
> **Author Response to Reviewer 1qwV (Part 1/2)**
>
> We are grateful for your positive review and valuable comments, and we hope our response fully resolves your concerns.
>
> >Q1. About the implementation details of CyberHost
>
> a) About Audio Feature: To fully leverage multi-scale audio features, we concatenate the hidden states from each encoder layer of the Wav2Vec model with the last hidden state, resulting in an audio feature vector of size 10752. To capture more temporal context, we use a temporal window of size 5 and construct an input audio feature of shape (5, 10752) for each video frame.
>
> b) Orthogonal Latents Bank:  We applied the Gram-Schmidt process to the learnable latents bank during each forward pass to ensure orthogonality.
>
> c) We directly use MSE loss as the loss function to supervise the mask prediction process. During training, the hand mask loss and face mask loss are weighted by 0.005 and 0.001 respectively in the total loss.
> Thank your very much for your reminder, all of these details are fully elaborated in Section D of the revised appendix
>
> >Q2. About the hand clarity score
>
> In the [video](https://cyberhostanony.github.io/static/videos/rebuttal/HAND_CLARITY_ABLATION.mp4), we show the impact on the generation results when different hand clarity score conditions are input. It can be seen that it will directly affect the generation quality and dynamic degree of gestures. As illustrated in Figure 13 of the paper, the hand clarity score reflects the clarity of the hands in the training data, thereby allowing the identification of training videos with varying quality of hand visuals. It enables the model to distinguish between high-quality and low-quality hand images during training, thereby guiding the model to generate clear hand visuals during inference by inputting a high hand clarity score. This is also supported by the ablation experiment in Table 1.
>
> >Q3. About the Pose-aligned Reference Feature
>
> 1. Effect Demonstration: In the [video](https://cyberhostanony.github.io/static/videos/rebuttal/POSE_ALIGN_ABLATION.mp4), we show the difference between the model with Pose-aligned Reference Feature and the model without Pose-aligned Reference Feature for the same examples. It is evident that for more complex initial poses, the Pose-aligned Reference Feature significantly improves the reasonableness of the generated results and reduces issues with synthesizing key parts like hands.
>
> 2. Pose-aligned Reference Feature: This feature helps the model better understand the reference image, as the generated result needs to maintain the ID information of the reference image, including the hands. Logically, the model must know the hand pose in the reference image to capture its texture. This means pose detection should be completed before image generation. However, for the pose detection task, the hand gestures seen in the current training data may not be sufficient to support highly robust detection, leading to "recognition errors" and subsequent generation errors. Using an expert model for detection can avoid this issue by providing additional information to prevent "recognition errors," thereby reducing "generation errors." As seen in the  video, without the Pose-aligned Reference Feature, the model confuses the reference image's arm with the clothing, resulting in an arm being stuck to the clothing in the generated output. The introduction of the Pose-aligned Reference Feature significantly improves such issues, enhancing the final generation quality.
>
> >Q4. About whether a one-stage model has an advantage over a two-stage model if an infinitely precise mesh detector exists.
>
> First, let's assume that this mesh predictor can capture details including muscle movements. We need to discuss the upper limits of one-stage and two-stage methods. For a one-stage framework like Cyberhost, the input is a single image and audio, and the output is a video. There are no strict limitations on the input format. With appropriate training data design, the single image can include a single person, multiple people, or even people and pets. The generated video can then depict singing, conversations, or interactions between people and animals. This high degree of input flexibility stems from its nature as a one-stage audio-to-video generation framework. In contrast, methods based on intermediate representations currently struggle to achieve such extensibility.
> From a feasibility standpoint, using high-quality mesh reconstruction methods like [1,2] involves significant computational time, space constraints, and equipment costs, which limit the scale of available training data. This makes it difficult to benefit from data scaling up.
>
> [1] Baradel* F, Armando M, Galaaoui S and et.al. Multi-HMR: Multi-Person Whole-Body Human Mesh Recovery in a Single Shot
>
> [2] Pavlakos G, Shan D,  Radosavovic I and et.al. Reconstructing Hands in 3D with Transformers

---

> ### Author Response · Authors · 2024-11-22
> **Author Response to Reviewer 1qwV (Part 2/2)**
>
> >Q5. About the spelling mistakes
>
> Thank you for your reminder. We have made the corrections in the corresponding section of the original text.
>
> >Q6. About the training details
>
> Thank you for reminder. We apologize for any lack of clarity in our paper. Both the video-driven body reenactment model and the multimodal-driven video generation model require separate training processes. During the training process for video-driven body reenactment, we need to replace the body movement map with the full-body skeleton map. For training multimodal-driven video generation model, we draw the skeleton map of both hands onto the body movement map. We have clarified these settings in Section D of the revised appendix.
>
> >Q7. About the effectiveness of predicting regional masks in all layers
>
> Thank you very much for your suggestions. They have been very enlightening, and we conducted verification experiments based on your direction. Firstly, we calculated the cross-entropy between the predicted hand mask from each layer and the hand region in the generated results on a batch of test data to measure the effectiveness of each attention block. For specific details, see Figure 10 in revised appendix.
> Based on the accuracy of the predicted masks from each layer, we created corresponding top-1, top-5, and top-10 experiments (where "top" refers to lower cross-entropy) to verify whether predicting masks and applying RAM in fewer layers could yield better results. Unfortunately, as shown in the table, better results were not achieved. We hypothesize that although the top-n layers have more precise hand structure prediction abilities, removing other layers with relatively poorer mask prediction capabilities still reduces the overall expressive power of the model.
>
> | Method | **FID** | **FVD** | **CSIM** | **SyncC** | **HKC** |
> | --- | --- | --- | --- | --- | --- |
> | top-1 | 39.22 | 664.3 | 0.421 | 6.271 | 0.853 |
> | top-5 | 36.44 | 594.0 | 0.470 | 6.512 | 0.869 |
> | top-10 | 33.88 | 573.1 | 0.489 | 6.549 | 0.874 |
> | All | 32.97 | 555.8 | 0.514 | 6.627 | 0.884 |
>
> **Summary**
>
> Following your comments, we added discussions on the implementation details, the effectiveness of hand clarity score, training details and the effectiveness of predicting regional masks in all layers in the revised appendix. We think this enhances the paper and better explains the method. Thank you again for your valuable comments and positive recommendation for our paper.

---

> > ### Comment · Reviewer_1qwV · 2024-11-22
> > **Response to authors' rebuttal**
> >
> > Thank the authors for their rebuttal.
> > Almost all of my concerns have been addressed. Perhaps one thing was not clearly expressed when I wrote question 3: what I meant was to predict the mask from one or several layers with the highest accuracy only, while still performing regional attention in all layers. I suggest that the authors consider conducting this in the future.
> > But still, the authors provide solid extended experiments, more implementation details, and reasonable explanations about the proposed questions. Taking the good results and novelty into consideration, I think this is really a good paper. Therefore, I decide to raise my rating to 10.

---

> ### Author Response · Authors · 2024-11-23
> **Author Response to Reviewer 1qwV**
>
> Thank you once again for your valuable suggestions and positive feedback.
>
> We apologize for any misunderstanding regarding the intent of question 3 earlier. To clarify, we predict independent region masks at various layers and utilize them to guide the corresponding regional attention for local feature learning. As illustrated in Figure 10, the middle and later layers of the U-Net achieve higher mask prediction accuracy. Due to the inherent feature dependency in the network's forward pass, the earlier layers are unable to leverage the masks predicted by the subsequent layers. Consequently, if masks are solely predicted at the later layers, it precludes the possibility of directly applying regional attention across all layers.
>
> However, inspired by this idea, we could also consider using the region mask predicted by the most accurate layer at timestep  **t+1** for all layers at timestep **t**.  Based on our [video visualizations](https://cyberhostanony.github.io/static/videos/rebuttal/HAND_MASK_VIS.mp4), the masks predicted at timestep **t+1** tend to be slightly less accurate than those at timestep  **t**. Therefore, implementing this optimization approach necessitates careful consideration of more complex comparative relationships.
>
> Nonetheless, the exploration of the effectiveness of different network layers for local feature learning is indeed a compelling topic. We intend to pursue further research on this aspect in the future.

---

### Official Review · Reviewer_XLFx · 2024-11-03

**Soundness:** 3
**Presentation:** 3
**Contribution:** 3
**Rating:** 6
**Confidence:** 4

**Summary:**

This paper proposes a one-stage audio-driven talking body video generation framework, addressing issues in half-body video generation such as blurred hand details, inconsistent identity, and unnatural motion. Specifically, it introduces a Region Attention Module (RAM) to enhance the quality of local regions. Additionally, it proposes a human-prior-guided condition to improve motion stability in generated videos. A new dataset was collected for experimentation, with results verifying the effectiveness of the proposed method and the improvements contributed by each component.

**Strengths:**

1.	The proposed method demonstrates a certain degree of generalization, allowing it to adapt to multiple tasks, such as video-driven generation or multimodal-driven generation, while also enabling open-set generation.
2.	Based on the experimental results, the proposed method surpasses both the baseline and state-of-the-art methods across multiple metrics.

**Weaknesses:**

1.	Although the proposed method achieves promising results overall, it introduces many components. As shown in Table 1, there are nine components, but the experiments lack in-depth analysis of these. For example, the impact of the size of the latent bank in RAM. The results of using alternatives in the Region Attention Module (RAM), such as not using spatial latents, were not examined. Additionally, the effect of not decoupling the latent bank into spatial and temporal latents—instead using a single 3D latent bank—was not investigated. Furthermore, it remains unclear what specific aspects of video information are captured by the spatial and temporal latents, lacking justification and explanation.
3.	The use of the Laplacian operator to compute the hand clarity score requires justification, as the rationale behind this choice is not explicitly discussed. Additionally, the influence of the hand clarity score on the experimental results is not demonstrated in the experiments. It is essential to clarify whether this score is necessary and how it contributes to the overall performance of the proposed method.
4.	The method [1] is also a one-stage audio-driven half-body video generation model, but this paper does not discuss or compare it.

5.	The dataset used in [2] was not employed in experiments for comparison with previous methods. Additionally, the beat consistency metric [3] was not reported in the experiments.

6.	Some typos, such as in line 313 feference -> reference

reference:

[1] Liu X, Wu Q, Zhou H, Du Y, Wu W, Lin D, Liu Z. Audio-driven co-speech gesture video generation.

[2] Qian S, Tu Z, Zhi Y, Liu W, Gao S. Speech drives templates: Co-speech gesture synthesis with learned templates.

[3] Li R, Yang S, Ross DA, Kanazawa A. Ai choreographer: Music conditioned 3d dance generation with aist++

**Questions:**

1.	When using full-body keypoints instead of the body movement map for video-driven generation, is it necessary to further fine-tune the entire model?
2.	How can hand pose templates be combined within the framework to achieve multimodal-driven generation? Does this process require fine-tuning the model?

---

> ### Author Response · Authors · 2024-11-22
> **Author Response to Reviewer XLFx (Part 1/2)**
>
> We are grateful for your review and valuable comments, and we hope our response fully resolves your concerns.
>
> >Q1. About the spatial and temporal latents
>
> Thank you very much for your suggestions.  Based on your suggestion, we conducted more ablation experiments on the Region Attention Module to verify the effectiveness of its structure. As shown in the table, eliminating the spatial bank significantly affects the quality of the generated videos, while the 3D Latents Bank, with a 55-fold increase in parameters under the current configuration, did not provide further performance gains. We hypothesize that the spatial and temporal latents bank, which more closely aligns with the behavior of U-Net by decoupling  the temporal attention from spatial attention,  consequently leads to more efficient learning of local features.  Additionally, due to time constraints, we only conducted ablation experiments on different sizes of the spatial latents bank in the RAM. **As shown in Figure 9** in the appendix of the revised version. The results indicate that increasing the size of the latents bank beyond the current configuration provides only marginal performance gains.
> To investigate the feature representations learned by the spatial latents and temporal latents, we visualized their respective feature maps used for the residuals. As shown in the [video](https://cyberhostanony.github.io/static/videos/rebuttal/LATENT_FEAT_VIS.mp4), the spatial latents focus more on all the texture regions of the hand, whereas the temporal latents emphasize the temporal smoothness information at the edges of the hand.
>
> | **Methods** | **FID** | **FVD** | **SyncC** | **HKC** | **HKV** |
> | --- | --- | --- | --- | --- | --- |
> | w/o Latents Bank | 37.80 | 643.9 | 6.384 | 0.859 | 21.35 |
> | w/o Spatial Latents | 36.23 | 612.7 | 6.362 | 0.864 | 22.15 |
> | w/o Temp. Latents | 36.75 | 624.1 | 6.468 | 0.870 | 19.44 |
> | 3D Latents Bank | 32.94 | 552.3 | 6.613 | 0.888 | 23.93 |
> | Spatial Latents + Temp. Latents | 32.97 | 555.8 | 6.627 | 0.884 | 24.73 |
>
> >Q2. About the detection of hand clarity score.
>
> (1) About the selection of hand clarity detection methods, first of all, we have tried to use the IQA expert models [1, 2] and the MLLM models [3, 4] to detect the blurriness of the crop hand image, but found that the effects are not good.  (2) In the [video](https://cyberhostanony.github.io/static/videos/rebuttal/HAND_CLARITY_ABLATION.mp4), we show the impact on the generation results when different hand clarity score conditions are input. It can be seen that it will directly affect the generation quality and dynamic degree of gestures. As illustrated in Figure 13 of the revised paper, the hand clarity score reflects the clarity of the hands in the training data, thereby allowing the identification of training videos with varying quality of hand visuals. It enables the model to distinguish between high-quality and low-quality hand images during training, thereby guiding the model to generate clear hand visuals during inference by inputting a high hand clarity score. This is also supported by the ablation experiment in Table 1.
>
> [1] Su S, Yan Q, Zhu Y, Zhang C, Ge Xin, Sun J, Zhang Y. Blindly Assess Image Quality in the Wild Guided by a Self-Adaptive Hyper Network
>
> [2] Wu H,  Zhang E, and Liao L, and et.al. Towards Explainable Video Quality Assessment: a Database and a Language-prompt Approach.
>
> [3] Wu H, Zhu  H,  Zhang Z and et.al. Towards Open-ended Visual Quality Comparison
>
> [4] Wu H, Zhang Z, Zhang W and et.al. Q-Align: Teaching LMMs for Visual Scoring via Discrete Text-Defined Levels
>
>
> >Q3. About the comparison with method ANGIE.
>
> Thank you for the reminder. Due to time constraints, we conducted a visual comparison with ANGIE[1] on PATS[4] dataset. In the [video](https://cyberhostanony.github.io/static/videos/rebuttal/CMP_ANGIE.mp4), we used Cyberhost to infer the demo video provided on their homepage. It is important to note that ANGIE is trained and tested simultaneously on the four individuals in the PATS dataset, whereas CyberHost focuses on driving open-set images. Despite not specifically training on the PATS dataset, our generated results exhibit natural motion patterns and significantly better video clarity.
> Although ANGIE establishes a unified framework to generate speaker image sequences driven by speech audio, the overall emphasis of this work is on the audio-to-motion generation aspect, utilizing a pre-existing image generator model from MRAA[5] for the visual synthesis. While our proposed CyberHost aims to optimize both motion generation and image generation simultaneously, providing a more unified algorithmic framework. We have included a discussion on the comparison with ANGIE[1] in the revised appendix.

---

> ### Author Response · Authors · 2024-11-22
> **Author Response to Reviewer XLFx (Part 2/2)**
>
> >Q4. About the dataset used in [2] and the beat consistency metric
>
> Thank you for your reminder. Due to time constraints, we conducted visual comparisons with previous methods on only a few examples from this dataset in the [video](https://cyberhostanony.github.io/static/videos/rebuttal/CMP_SPEECH2GESTURE.mp4). Despite these methods being specifically designed for video clips of a few characters, whereas our approach targets audio-driven video generation for open-set images, we still achieved relatively natural driving results.
> Besides, in the audio-driven scenario, we supplemented the beat consistency metric and compared it with the baseline methods. Since our approach does not directly generate keypoints, we uniformly used DWPose to detect 2D keypoints of the generated videos and calculated the beat consistency metric using the speed of these 2D keypoints for comparison. However, we found that this metric has limited discriminative power in evaluating performance for upper-body speaking scenarios; even Ground Truth videos did not achieve a relatively high score. We hypothesize that this may be due to our more open data collection sources, where the recorded characters do not intentionally exhibit intense emotional movements.
>
> Please note that we have included a discussion on the comparison results on the Speech2Gesture[6] dataset in the revised appendix.
>
> | Methods | Beat Consistency Score |
> | --- | --- |
> | DiffTED | 0.216 |
> | DiffGest.+MimicMo. | 0.205 |
> | CyberHost (A2V-B) | 0.207 |
> | Ground Truth | 0.192 |
> | Vlogger * | 0.223 |
> | CyberHost (A2V-B) * | 0.231 |
>
> ∗ denotes evaluate on vlogger test set.
>
> >Q5. About the training particulars of video/multimodality-driven generation
>
> We apologize that we did not describe this clearly. For the new driving method, the model needs to be retrained. We have supplemented the detailed settings and training details in the revised version.
>
> >Q6. About the typos and writing
>
> Thank you very much for the reminder. We have corrected these issues in the revised version and also carefully revised the full text.
>
> **Summary**
>
> Following your comments, we added discussions on the structural effectiveness of RAM, the effectiveness of hand clarity score, the comparison with ANGIE and the discussion of comparison results on Speech2Gesture[6] dataset  in the revised appendix. We think this enhances the paper and better explains the method. Thank you again for your valuable comments for our paper.
>
> [1] Liu X, Wu Q, Zhou H, Du Y, Wu W, Lin D, Liu Z. Audio-driven co-speech gesture video generation.
>
> [2] Qian S, Tu Z, Zhi Y, Liu W, Gao S. Speech drives templates: Co-speech gesture synthesis with learned templates.
>
> [3] Li R, Yang S, Ross DA, Kanazawa A. Ai choreographer: Music conditioned 3d dance generation with aist++
>
> [4] Ahuja C, and Lee D, Won N and et.al. Gestures Left Behind: Learning Relationships between Spoken Language and Freeform Gestures
>
> [5] Siarohin A and Woodford O and Ren J and et.al. Motion Representations for Articulated Animation
>
> [6] Shiry G, Amir B, Gefen K and et.al. Learning Individual Styles of Conversational Gestures

---

> > ### Comment · Reviewer_XLFx · 2024-11-24
> >
> > Thank you for your patient responses. Your answers have addressed most of my concerns. However, I believe the paper could benefit from more detailed analysis of the components listed in Table 1. For example, what are the results when using only the RAM module without other components? Does the use of the body movement map restrict the speaker's movement or the camera's mobility? If the ID-descriptor is removed, does it primarily affect ID consistency, or are there other issues? Nevertheless, I will raise my score to a supportive level.

---

> > > ### Author Response · Authors · 2024-11-25
> > > **Author Response to Reviewer XLFx**
> > >
> > > We sincerely appreciate your valuable suggestions and recommendations. We are pleased to see that our previous responses addressed your concerns. We apologize for any confusion caused by our earlier lack of clarity. In light of your new questions, we provide the following detailed explanations to hopefully eliminate any remaining concerns.
> > >
> > > We validated the effectiveness of each component of the RAM module through quantitative results presented in Table 1, Table 5, and Figure 9. Additionally, we provided visual comparisons to demonstrate the effectiveness of the Latents Bank ([video](https://cyberhostanony.github.io/static/videos/rebuttal/LATENT_ID_ABLATION.mp4)), ID Descriptor ([video](https://cyberhostanony.github.io/static/videos/rebuttal/LATENT_ID_ABLATION.mp4)), and Temporal Latents (video in supplementary material, timestamp 1:40). Specifically, as shown in the [video](https://cyberhostanony.github.io/static/videos/rebuttal/LATENT_ID_ABLATION.mp4), the removal of the ID Descriptor leads to inconsistencies in facial features and skin tone. The quantitative results in Table 1 also indicate that the ID descriptor primarily affects the ID-related CSIM metric, while it has a minimal impact on hand quality metrics such as HKC.
> > >
> > > In the lower half of Table 1, we quantitatively analyzed the ablation of each component of the Human-Prior-Guided Conditions (i.e., components other than the RAM module). Additionally, visual comparisons are provided to illustrate the effects of the Pose-Aligned Reference Feature ([video](https://cyberhostanony.github.io/static/videos/rebuttal/POSE_ALIGN_ABLATION.mp4)) and Hand Clarity Score ([video](https://cyberhostanony.github.io/static/videos/rebuttal/HAND_CLARITY_ABLATION.mp4)). For the Body Movement Map, its removal results in unstable absolute displacement of the character (see  the video in supplementary material, timestamp 2:00). We applied an augmentation to the body movement map during training, preventing it from completely restricting body movements, thereby allowing for slight motions such as turning, as demonstrated in the [video](https://cyberhostanony.github.io/static/videos/A2V_CMP_CROP.mp4) at timestamp 0:14. It is true that this constraint limits the model's ability to generate large body movements and camera shifts. However, this is intentional by design, as our primary focus is on talking body generation, specifically the facial and hand movements related to speech. By concentrating on optimizing these aspects, we aim to enhance the expressiveness and overall quality of the generated videos.
> > >
> > > Overall, for the various specific components of the RAM module, we provided detailed quantitative comparisons in Table 1, Table 5, and Figure 9, as well as in multiple visual comparison videos ([video](https://cyberhostanony.github.io/static/videos/rebuttal/LATENT_ID_ABLATION.mp4) and the video in supplementary material), to demonstrate their effectiveness. For the specific components of the Human-Prior-Guided Conditions, quantitative and qualitative analyses are provided in Table 1, [video1](https://cyberhostanony.github.io/static/videos/rebuttal/POSE_ALIGN_ABLATION.mp4), [video2](https://cyberhostanony.github.io/static/videos/rebuttal/HAND_CLARITY_ABLATION.mp4) and the video in supplementary material. These detailed experimental analyses support the effectiveness of the proposed module and demonstrate that the overall framework achieves satisfactory final synthesis results.
> > >
> > >
> > > Finally, we thank you for your valuable suggestions. We are more than willing to answer any further questions you may have. We believe these queries contribute significantly to the completeness and robustness of our method. We hope our response fully addresses your concerns.

---

### Official Review · Reviewer_r6Nr · 2024-11-04

**Soundness:** 3
**Presentation:** 2
**Contribution:** 3
**Rating:** 6
**Confidence:** 4

**Summary:**

The paper introduces an end-to-end audio-driven human animation framework, which is designed to generate realistic and natural upper body human videos from a single image and control signals such as audio, ensuring hand integrity, identity consistency, and natural motion.

**Strengths:**

1. Cyberhost can generate cospeech videos with very natural motions and clear hand/body structures.
2. It employs various control training methods, including codebook , hand clarity, pose-aligned reference, and also key point supervision. Experimental results indicate that these methods effectively enhance the clarity of hands and the correctness of body structures in the generated objects.

**Weaknesses:**

1. The generated videos exhibit insufficient facial clarity and detail, resulting in a noticeable discrepancy between the generated object and the characteristic features of the person in the reference image.
2. Unlike the codebook in VQ-VAE, which is specifically used for the reconstruction of designated features, the codebook in Cyberhost lacks supervisory signals during training, making it unable to ensure that the codebook effectively guides the model to generate correct hand shapes and facial features.
3. It would be good to visualize the ablation study for the two main contribution components: “Motion codebook” and "ID Descriptor".

**Questions:**

1. There are issues with the injection of the codebook during inference, and the paper does not clearly explain how to accurately detect the hand position from the noisy latent space when the timestep corresponds to a higher noise level.

---

> ### Author Response · Authors · 2024-11-22
> **Author Response to Reviewer r6Nr**
>
> We are grateful for your review and valuable comments, and we hope our response fully resolves your concerns.
>
> >Q1. About the clarity and detail of generated results.
>
> Thank you for the reminder. Although Cyberhost cannot achieve the same clarity as the reference images, it has reached the best performance in terms of image quality metrics such as SSIM，PSNR and FID in Tables 1 at the current resolution. Additionally, it leads in video quality metrics like FVD. This paper focuses on building a one-stage audio-driven talking body video generation diffusion framework. The experimental results validate the feasibility of this approach and show significant advantages in motion naturalness and video quality compared to existing methods.
> To further improve the results, some data cleaning strategies can be adopted. We can enhance output quality by removing low-resolution data and further increasing resolution. Training with multi-resolution data in stages can also help. Enhancing fine detail generation is a common issue, and we aim to address this more effectively in future work based on the Cyberhost framework. In the [video](https://cyberhostanony.github.io/static/videos/rebuttal/HDR_RES_480P.mp4), we also present several test results from our high-resolution model version (640x480), which exhibit superior performance in maintaining identity and preserving details.
> This result is only to demonstrate that the proposed cyberhost can generate finer image quality, and its potential can still be enhanced by high-quality data, not limited by the framework itself.
>
> >Q2.  Regarding whether the codebook can effectively guide the model to generate correct hand shapes and facial features.
>
> First, we apologize for any unclear representations in our method. We assume that the "codebook" mentioned by the reviewer refers to our designed latents bank. We address this question from two perspectives: (1) Method Design: During training and testing, we use predicted regional masks to focus region latents on specific areas like hands and face. The [video](https://cyberhostanony.github.io/static/videos/rebuttal/HAND_MASK_VIS.mp4) shows that predicted masks accurately capture these regions, ensuring the region latents functions correctly. (2) Visualization Analysis: In the [video](https://cyberhostanony.github.io/static/videos/rebuttal/HAND_MASK_VIS.mp4), by removing face and hand region latents attentions on a trained full model during inference, we observed that the corresponding regions were significantly affected while other parts remained largely unchanged. This, along with the quantitative and qualitative results in Table 1 and the supplementary video, demonstrates that region latents are crucial for generating local areas.
>
> >Q3. About the visualization of the ablation study for the "Motion Codebook" and "ID Descriptor".
>
> In the [video](https://cyberhostanony.github.io/static/videos/rebuttal/LATENT_ID_ABLATION.mp4), we provide the visualization of the ablation study for the "latent bank" (which may refer to the motion codebook) and the "ID Descriptor".
>
> >Q4, About the detection of the hand position during inference
>
> Although we only train the hand mask prediction at low timesteps, we found that it generalizes well to different timesteps during inference. In the [video](https://cyberhostanony.github.io/static/videos/rebuttal/HAND_MASK_VIS.mp4), we provide a visualization of the predicted region masks. It shows that in the early stages of denoising, when the hand generation is unclear, the predicted hand mask is also rough. As the denoising steps increase, the prediction becomes more stable, which should be sufficient for the region latents attention to be effective.
>
> **Summary**
>
> Thank you for your suggestions. We will include the ablation results of the Latents Bank & ID Descriptor and the visualization of the predicted hand mask on our project website in the future. We think this enhances the paper and better explains the method. Thank you again for your valuable comments for our paper.

---

> ### Author Response · Authors · 2024-11-25
> **Author Response to Reviewer r6Nr**
>
> Thank you once again for your valuable feedback.
>
> We have included more experimental results and visualizations based on your comments. We hope you find the response satisfactory. As the discussion phase nears its end, we look forward to any additional feedback to ensure all concerns are addressed.
>
> Best regards

---

### Official Review · Reviewer_R5UP · 2024-11-05

**Soundness:** 3
**Presentation:** 3
**Contribution:** 4
**Rating:** 8
**Confidence:** 4

**Summary:**

The paper introduces CyberHost, an innovative one-stage audio-driven framework for generating talking body animations, addressing common issues such as hand integrity, identity consistency, and natural motion. Unlike multi-stage methods using intermediate representations like poses or meshes, CyberHost works end-to-end and supports zero-shot generation.

Key innovations like Region Attention Module and the usage of Human-Prior-Guided Conditions are proposed to improve the generation quality of local human regions and to address the motion uncertainty problem.

Experiments show CyberHost outperforms existing methods both qualitatively and quantitatively and works well in audio-driven, video-driven, and hybrid scenarios.

**Strengths:**

1. CyberHost introduces the first one-stage approach for audio-driven talking body generation, avoiding the complexity and inefficiencies of multi-stage systems that rely on intermediate representations.

2. The proposed Region Attention Module component effectively enhances critical areas such as hands and faces, improving the quality of local details and maintaining identity consistency.

3. By integrating motion constraints and structural priors via human-prior-guided conditions, the model mitigates the challenge of motion uncertainty, resulting in more stable and natural body animations.

4. The qualitative results in the supplementary materials are impressive. Also, compared to the previous state-of-the-art audio-driven half-body generation method, VLOGGER, CyberHost produces visibly superior results.

5. The paper is well-written and clearly presents its objectives, methodology, and findings.

**Weaknesses:**

1. Detailed Failure Analysis: The paper would benefit from a discussion of failure cases or limitations where CyberHost struggles, such as specific types of input audio or complex poses. This would provide a more balanced view of the model's capabilities.

2. Scalability to Full-Body Generation: The paper focuses on half-body animation, but it does not discuss how well the architecture scales to full-body animation or if there are significant challenges in extending the framework.

3. Lack of User Study for Subjective Evaluation: The paper does not include user studies or subjective evaluations to gather feedback on the perceived naturalness and quality of the generated videos. Such evaluations would provide valuable insights into how well the model meets human expectations for lifelike animation.

**Questions:**

1. Will the dataset used for training and evaluation be made publicly available? This would be valuable for reproducibility and further research by other researchers.

2. Failure Cases: What are the known limitations or specific scenarios where CyberHost struggles? Highlighting these would give a more complete picture of the model’s strengths and areas for improvement.

3. Full-Body Animation Scalability: Can the model be adapted for full-body animation, and if so, are there significant challenges or limitations to scaling up from half-body to full-body scenarios?

4. User Study Inclusion: Could authors conduct user studies for subjective evaluations to gather human feedback on the perceived quality of the generated videos?

5. DiffGesture Baseline: In the experiments section, the authors mentioned that they trained DiffGesture on the collected dataset, how did the authors get the SMPLX annotations for the collected dataset? It would also be good if the authors can quantitatively and qualitatively assess the generated SMPLX quality of the trained DiffGesture.

---

> ### Author Response · Authors · 2024-11-22
> **Author Response to Reviewer R5UP (Part 1/2)**
>
> We are grateful for your positive review and valuable comments, and we hope our response fully resolves your concerns.
>
> >Q1: About the discussion of failure cases or limitations.
>
> Thank you very much for your suggestion. We have included an analysis of the current model's limitations in the [video](https://cyberhostanony.github.io/static/videos/rebuttal/FAILURE_CASE.mp4). CyberHost still cannot handle the following situations:
>
> For the input image: Exaggerated body proportions in cartoon images often result in unrealistic outputs; Complex backgrounds resembling hand colors, such as skin-toned backgrounds, are prone to generating errors; Additionally, challenging human poses, such as raising both hands above the head, complicate the generation of natural motion.
>
> For the input audio: scenarios with complex background sounds and varying tones, such as singing, make it difficult to generate accurate lip movements and naturally synchronized motions.
>
> Possible approaches to address these issues include: 1) Collecting a more diverse dataset of character videos, including anime scenes and data featuring complex human poses. 2) Replacing Wav2Vec with a more advanced audio feature extractor and implementing background noise reduction strategies.  We have also provided a more detailed discussion in the appendix of the revised version.
>
> >Q2. About the scalability to full-body generation
>
> Thank you for your reminder. Full-body scenes present significant challenges, which we plan to address in future work. In the [video](https://cyberhostanony.github.io/static/videos/rebuttal/FULL_BODY.mp4), we have provided examples of the current results in full-body scenes. From a methodological perspective, Cyberhost can be expanded to full-body scenarios. But right now, there are two main reasons why the performance isn't that good: (1) The VAE  struggle to effectively model faces and hands due to their smaller proportion in full-body frames. This can be fix by increasing the training resolution and model capacity. (2) The current training data mostly focuses on upper-body scenes, so we need to collect and train on full-body scene data. Full-body scenarios are challenging but also very useful, and this is exactly what we hope to address in our future work.
>
> >Q3. About the user study for subjective evaluation
>
> Thank you for the reminder. Due to time constraints, we conducted a user study with five independent, professional evaluators. Below is a summary of the questionnaire results. We used the MOS (Mean Opinion Score) rating method, with specific criteria detailed in the appendix of the revised version. As can be seen, Cyberhost shows performance advantages across multiple dimensions compared to the baseline methods. Evaluation results involving more participants will be included in the final version.
>
> | **Methods** | **ID Consistency** | **Motion Naturalness** | **Video Quality** | **Lip-sync** |
> | --- | --- | --- | --- | --- |
> | DiffTED | 1.54 | 1.52 | 1.92 | 1.00 |
> | DiffGest.+MimicMo. | 3.28 | 2.56 | 3.32 | 2.34 |
> | CyberHost (A2V-B) | 4.40 | 4.28 | 4.10 | 4.64 |
> | Ground Truth | 4.94 | 4.72 | 4.66 | 4.88 |
>
> | **Methods** | **ID Consistency** | **Motion Naturalness** | **Video Quality** | **Lip-sync** |
> | --- | --- | --- | --- | --- |
> | Vlogger * | 4.10 | 3.54 | 3.26 | 2.40 |
> | CyberHost (A2V-B) * | 4.62 | 4.12 | 4.52 | 4.70 |
>
> \* denotes evaluate on vlogger test set.
>
> >Q4. About the release of the training dataset
>
> Thank you for your suggestion. At this time, we are indeed unable to provide a definite plan for making the dataset publicly available, as this decision requires further consideration and internal data review processes. However, we have included a detailed data filtering process and data composition analysis in the revised version, and we hope this will be helpful.

---

> > ### Author Response · Authors · 2024-11-22
> > **Author Response to Reviewer R5UP (Part 2/2)**
> >
> > >Q5. About the SMPLX annotations and SMPLX quality of the trained DiffGesture.
> >
> > Thank your for your interest.
> >
> > 1) For SMPLX annotations: We utilize Multi-HMR[1] and HaMeR[2] for the reconstruction of human bodies and hands in the video, respectively. Subsequently, we employ a 2D keypoint fitting postprocess and temporal smoothing to further enhance the generation quality of SMPLX annotations.
> >
> > 2) For the generated SMPLX quality of the trained DiffGesture:
> > We have observed that training on a small number of IDs (fewer than 20 individuals) with long-duration videos (2-3 hours) and testing on a close-set ID results in better performance for audio-to-motion generation. However, the generated actions and poses are relatively monotonous and cannot adapt to the diverse initial poses in an open set.
> >  However, in our application scenario, training is required on a large number of IDs (10k) with short-duration videos (2-10 seconds) and testing is requred on a open-set ID, which results in less satisfactory audio2motion generation performance.  In the [video](https://cyberhostanony.github.io/static/videos/rebuttal/DIFFGES_VIS.mp4), we have provided test results for our audio2motion model trained in both scenarios. To ensure a fair comparison with our proposed one-stage audio-driven method, we restricted the audio2motion and motion2video modules of the two-stage baseline to open-set usage scenarios.
> > Regrettably, due to time constraints, we were unable to complete SMPLX annotation and the corresponding training and testing on the TED dataset, and thus cannot provide quantitative metrics.
> >
> > [1] Baradel* F, Armando M, Galaaoui S and et.al. Multi-HMR: Multi-Person Whole-Body Human Mesh Recovery in a Single Shot
> >
> > [2] Pavlakos G, Shan D,  Radosavovic I and et.al. Reconstructing Hands in 3D with Transformers
> >
> > **Summary**
> >
> > Following your comments, we added discussions on failure cases & limitation, full-body generation and subjective evaluation in the revised appendix. We think this enhances the paper and better explains the method. Thank you again for your valuable comments and positive recommendation for our paper.

---

> > > ### Comment · Reviewer_R5UP · 2024-12-02
> > >
> > > Thank the authors for their rebuttal. Most of my concerns were addressed. I think this is a good paper, I will keep my score for acceptance.

---

### Author Response · Authors · 2024-11-24
**General Response to All Reviewers**

We sincerely appreciate the positive feedback from the reviewers on our proposed method. This encouragement is highly valuable to us. In our responses, we have diligently addressed the reviewers' comments by providing additional experimental results and discussions. We believe these enhancements make our paper more comprehensive.

---

**Effectiveness of Our Method**
* We have conducted additional ablation studies to validate the effectiveness of the current design of the spatial and temporal latents bank. Additionally, we have provided visualization analyses  ([video1](https://cyberhostanony.github.io/static/videos/rebuttal/HAND_MASK_VIS.mp4), [video2](https://cyberhostanony.github.io/static/videos/rebuttal/LATENT_FEAT_VIS.mp4)) to verify their respective roles in local feature learning.
* We have explained the motivation for using the hand clarity score and provided visualizations ([video](https://cyberhostanony.github.io/static/videos/rebuttal/HAND_CLARITY_ABLATION.mp4)) to demonstrate its effectiveness. Additionally, we have also conducted visual ablation studies for the pose-aligned reference feature ([video](https://cyberhostanony.github.io/static/videos/rebuttal/POSE_ALIGN_ABLATION.mp4)), latents bank and ID descriptor ([video](https://cyberhostanony.github.io/static/videos/rebuttal/LATENT_ID_ABLATION.mp4)).
* We showcased the video generation capabilities of CyberHost at a resolution of 640x480 ([video](https://cyberhostanony.github.io/static/videos/rebuttal/HDR_RES_480P.mp4)), illustrating its leading quality in the field of talking body generation.
* We discussed the advantages of our one-stage method compared to a two-stage method and provided more implementation details of our DiffGesture Baseline ([video](https://cyberhostanony.github.io/static/videos/rebuttal/DIFFGES_VIS.mp4)).

**More Experimental Results**
* We provided the subjective evaluation results (MOS) and supplemented the MOS criteria in the revised appendix.
* We have included visual comparisons with more audio-driven methods on public datasets ([video1](https://cyberhostanony.github.io/static/videos/rebuttal/CMP_ANGIE.mp4), [video2](https://cyberhostanony.github.io/static/videos/rebuttal/CMP_SPEECH2GESTURE.mp4)) , and also comparative results for the Consistency Beat Score metric with baseline methods.
* We designed experiments to demonstrate the effectiveness of employing regional mask predictor and RAM module in different layers.

**Implementation Details**
* We have enriched the appendix with more implementation details, including model initialization, loss functions, latents bank implementation and training details.

**Limitations and Future Work**
* We have discussed some of CyberHost’s current failure cases and limitations from the perspectives of both input image and input audio ([video](https://cyberhostanony.github.io/static/videos/rebuttal/FAILURE_CASE.mp4)). We explored the scalability and challenges of applying our method to full-body generation ([video](https://cyberhostanony.github.io/static/videos/rebuttal/FULL_BODY.mp4)) and provided insights into constructing a review system in the future work.

Based on the reviewers’ suggestions, we have also modified the paper, supplementing necessary implementation details, clarifying ambiguous expressions, and correcting some writing errors to make the paper easier to understand.

---

We sincerely appreciate the valuable suggestions and positive recommendations from the reviewers. We believe that the aforementioned additions, significantly enhance the completeness and value of the paper. We have meticulously addressed the main concerns and provided detailed responses to each reviewer. We hope that you find our responses satisfactory and would be grateful to receive your feedback on our answers to the reviews.

Sincerely
Authors of Paper 4230

---

### Meta-Review · Area_Chair_SdwH · 2024-12-23

**Metareview:**

This paper proposes a one-stage diffusion model for audio-driven talking upper-body generation. The main innovations of this approach are a Region Attention Module and using Human-Prior-Guided Conditions to improve the quality of the local hand and face regions, and to reduce the ambiguity of driving body motion with the weaker audio signal, respectively. The authors compare their approach to various state-of-the-art approaches for upper body and face animation and show superior quality, both quantitatively and qualitatively.

The work proposes the first one-stage diffusion model for audio-driven talking body generation with high quality appearance of face and hands, and natural motion. This is an important problem to address towards enabling the practical use of human animation technology in real-world applications. Floatings heads just don't cut it and limit realism. It also demonstrates clearly superior quality versus the current state of the art and hence advances the field of human animation forward.

**Additional Comments On Reviewer Discussion:**

Five reviewers provided the final scores of 8, 6, 6, 10, 8 for this work. The reviewers were generally very positive about the innovative and elegant design of the proposed method and impressed with the visual quality of the results, showcasing a significant advancement in the field. The reviewers initially raised several concerns about the clarify of the method; requested additional experimental results and comparisons to the state of the art; and additional ablations of the current method. The authors' detailed responses and additional results during the rebuttal phase mostly satisfied all reviewers' concerns and they all increased their scores towards a positive one. In addition, several reviewers championed this work with scores of 8, 8 and 10.

The AC concurs and recommends acceptance.

---

### Decision · Program_Chairs · 2025-01-22

Accept (Oral)